# In vivo imaging of mitochondrial DNA mutations using an integrated nano Cas12a sensor

Yanan Li [1], Yonghua Wu[1], Ru Xu[1], Jialing Guo[1], Fenglei Quan[1], Yongyuan Zhang[1], Di Huang[1], Yiran Pei[1], Hua Gao[1], Wei Liu[1], Junjie Liu [1], Zhenzhong Zhang[1] ✉, Ruijie Deng [2] ✉, Jinjin Shi [1] ✉ & Kaixiang Zhang [1] ✉

Mutations in mitochondrial DNA (mtDNA) play critical roles in many human diseases. In vivo visualization of cells bearing mtDNA mutations is important for resolving the complexity of these diseases, which remains challenging. Here we develop an integrated nano Cas12a sensor (InCasor) and show its utility for efficient imaging of mtDNA mutations in live cells and tumor-bearing mouse models. We co-deliver Cas12a/crRNA, fluorophore-quencher reporters and $Mg^{2+}$ into mitochondria. This process enables the activation of Cas12a's trans-cleavage by targeting mtDNA, which efficiently cleave reporters to generate fluorescent signals for robustly sensing and reporting single-nucleotide variations (SNVs) in cells. Since engineered crRNA significantly increase Cas12a's sensitivity to mismatches in mtDNA, we can identify tumor tissue and metastases by visualizing cells with mutant mtDNAs in vivo using InCasor. This CRISPR imaging nanoprobe holds potential for applications in mtDNA mutation-related basic research, diagnostics and gene therapies.

Inherited or acquired mutations in mitochondrial DNA (mtDNA) result in a variety of human metabolic diseases[1,2]. mtDNA mutations are mainly caused by the high level of reactive oxygen species produced during oxidative phosphorylation, and the lower fidelity of DNA replication in mitochondria[3]. Due to their elevated mutation rate and constrained repair capacity, mtDNAs typically exist in a heteroplasmic state with a high prevalence of pathogenic mutations[4,5]. Moreover, acquired mutations would accumulate during life and finally contribute to various aging-related diseases when reaching a threshold[6,7]. Despite their importance, the difficulty of identifying mtDNA mutations at high spatial resolution in single cells and animal models has limited the study on such biological processes.

In situ rolling circle amplification (RCA) methods have been applied to visualize the coexistence of wild-type and mutant mtDNAs in fixed single cells or tissue slices[8,9]. By genotyping individual mtDNAs with high spatial resolution, these techniques enable investigations into the organization of mtDNA, thus providing fundamental insights into the development of mitochondrial disease[10]. However, due to the participation of numerous enzymatic stages, in situ RCA cannot be used for live cell or in vivo study[11]. Since mtDNA heterogeneity develops over time, it is necessary to develop probes for monitoring disease-associated mtDNA mutations in vivo, which, to the best of our knowledge, has not been developed.

CRISPR systems developed in recent years have provided powerful molecular recognition tools for tracking genome dynamics in live cells[12–17]. However, most of them used dCas9[14] for imaging of genomic loci dynamics in live cells[16,18], while single-nucleotide variation (SNV) imaging remains a challenge[19]. To solve this issue, we tend to focus on other subtypes of CRISPR systems. Among them, CRISPR/Cas12a is an RNA-guided enzyme, which can bind and cut target DNA while

[1]School of Pharmaceutical Sciences, Key Laboratory of Targeting Therapy and Diagnosis for Critical Diseases, Collaborative Innovation Center of New Drug Research and Safety Evaluation, State Key Laboratory of Esophageal Cancer Prevention & Treatment, Zhengzhou University, Zhengzhou 450001, China. [2]College of Biomass Science and Engineering, Sichuan University, Chengdu 610065, China. ✉e-mail: zhangzhenzhong@zzu.edu.cn; drj17@scu.edu.cn; shijinyxy@zzu.edu.cn; zhangkx@zzu.edu.cn

activating trans-cleavage activity for indiscriminate degradation of nearby single-stranded DNA (ssDNA) molecules[20–22]. This capability has been exploited for building efficient nucleic acid diagnostic tools, such as SHERLOCK[23] and DETECTR[20], for rapid detection of specific pathogens in clinical samples, including SARS-CoV-2[24].

Recently, CRISPR/Cas12a have been used to monitor nucleic acids, small molecules, ions and other biomarkers in live cells and in vivo[25–27]. For example, Song et al. introduce a CRISPR/aptamer-based sensor embedded in an organic framework for in vivo ATP imaging[26]. Li and his team demonstrated a CRISPR/Cas12a biosensor for in vivo spatiotemporally imaging of mRNA[28]. However, for mtDNA mutation detection, the Cas12a probes need to be efficiently delivered into mitochondria and the SNV recognition specificity of Cas12a needs further improved. In addition, to achieve sensitive

detection of mtDNA mutation, the collateral trans-cleavage activity of Cas12a in live cells need to be fully activated.

In this work, we design an integrated nano Cas12a sensor (InCasor), which is able to efficiently deliver Cas12a into mitochondria in live cells for recognizing target mtDNA, which can activate the trans-cleavage activity of Cas12a to cleave the co-delivered circular reporter (CLR), generating robust fluorescence signals to report mtDNA mutations (illustrated in Fig. 1a). By crRNA engineering, we can specifically identify SNVs in target mtDNA. Besides, by supplying sufficient amount of $Mg^{2+}$ into the cellular environment, the collateral trans-cleavage activity of Cas12a can be substantially enhanced, amplifying the detection signal within live cells. We demonstrate that InCasor is able to identify mtDNA mutations in vivo, thereby showing high potential for various biological and biomedical applications.

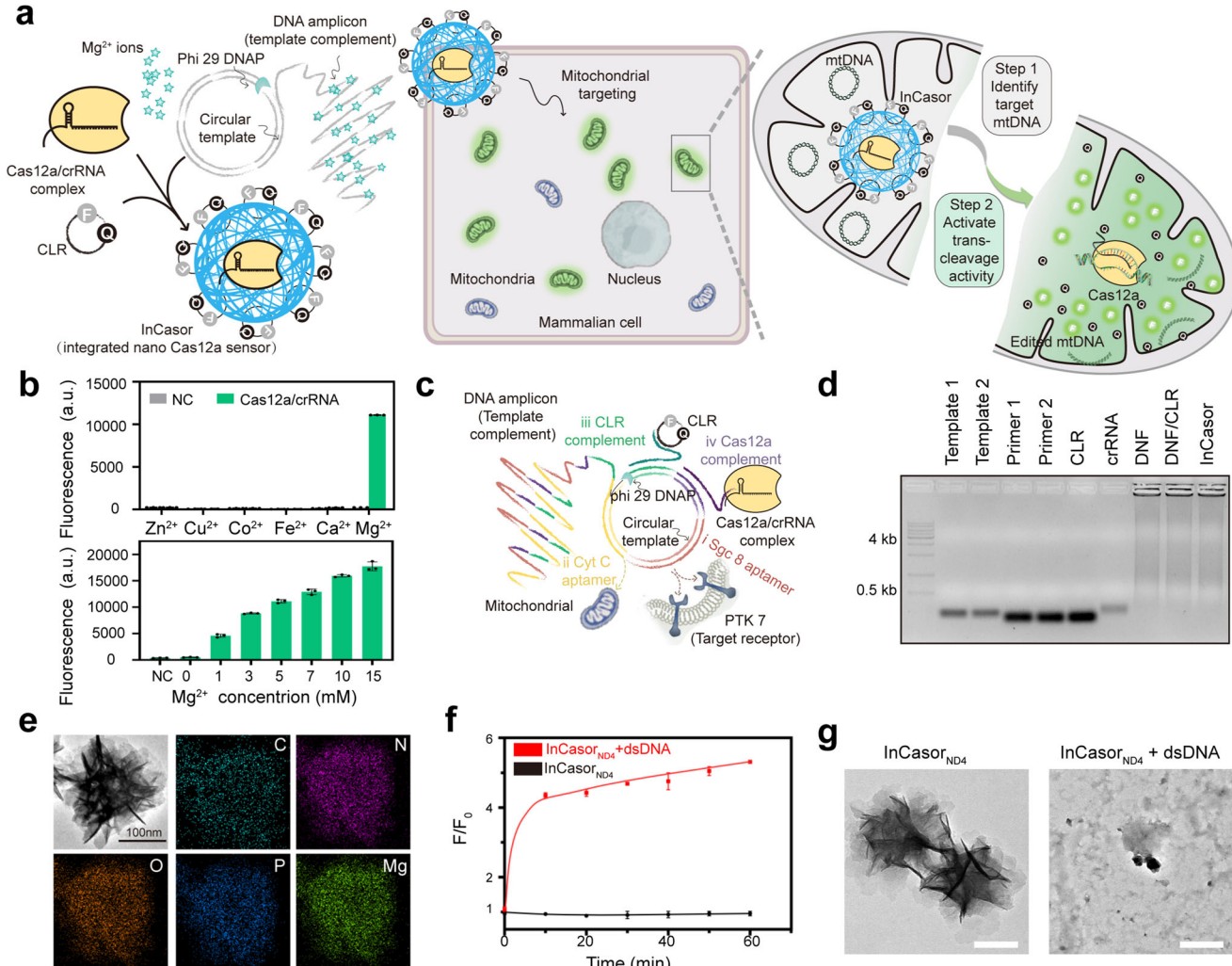

**Fig. 1 | Design and characterization of InCasor. a** Schematic of the integrated nano Cas12a sensor (InCasor) consisting of a DNA/$Mg^{2+}$ hybrid nanoflower (DNF), Cas12a/crRNA and circular reporter (CLR) for mtDNA imaging in live cells. **b** Analysis of the divalent cation preference of Cas12a/crRNA using a fluorophore-quencher single-stranded DNA cleavage assay. $n = 3$, data show mean ± SD. **c** Design of multielement-encoded DNA template used for rolling circle amplification (RCA) to generate DNA/$Mg^{2+}$ hybrid nanoflower (DNF). The complementary DNA template features 4 main elements: i) cell-targeting aptamer (Sgc8); ii) mitochondria-targeting aptamer (Cytochrome C aptamer); iii) circular reporter (CLR) binding element; iv) Cas12a/crRNA binding element. Phi 29 DNA polymerase produces long ssDNA with repeated copies of the complementary sequence of the template, which could envelop magnesium pyrophosphate generated during DNA

synthesis to form a nanoflower structure. **d** Agarose gel electrophoresis analysis of InCasor preparation steps: Lane 1: DNA ladder, Lane 2: Template 1 (encoding Sgc8 aptamer and binding site for Cas12a/crRNA complex), Lane 3: Template 2 (encoding Cyt C apt and binding site for CLR, Lane 4: Primer 1, Lane 5: Primer 2, Lane 6: CLR, Lane 7: crRNA, Lane 8: DNF, Lane 9: DNF/CLR, Lane 10: InCasor. **e** The scanning transmission electron microscopy (STEM) and energy dispersive spectroscopy (EDS) mapping of InCasor probe. Scale bar: 100 nm. **f, g** InCasor_ND4 can recognize target mtDNA in vitro to generate a fluorescence signal (**f**) and be self-degraded (**g**). The experiment was independently repeated three times with similar results. Data are presented as mean ± SD. Source data from (**b**, **d**, and **f**) are provided as a Source Data file.

## Results

### Design and characterization of InCasor

As Cas12a is a metal-dependent endonuclease[29], we first tested the effects of a range of divalent metal cations ($Zn^{2+}$, $Cu^{2+}$, $Co^{2+}$, $Fe^{2+}$, $Ca^{2+}$ and $Mg^{2+}$) on the activation of ssDNA cleavage activity of LbCas12a. We found that $Mg^{2+}$ supported the highest trans-cleavage activity for LbCas12a (Fig. 1b, top) and the fluorescence signal increased when $Mg^{2+}$ concentrations increased (Fig. 1b, bottom). Our results are consistent with literature reports that $Mg^{2+}$ is crucial for activating its trans-cleavage activity[30]. Therefore, we chose 10 mM $Mg^{2+}$ concentration for subsequent experiments.

After demonstration of the crucial role of $Mg^{2+}$ for activating trans-cleavage of Cas12a, we need to design a nanocarrier that can efficiently transport Cas12a/crRNA complex, fluorophore-quencher reporter, and a large amount of $Mg^{2+}$ into mitochondria to develop an integrated nanoprobe for in vivo mtDNA mutation imaging (Fig. 1a). Based on our previous research, we synthesized a DNA-$Mg^{2+}$ hybridized nanoflower (DNF) by rolling circle amplification (RCA) to achieve this goal[31]. The DNF encoded 2 DNA aptamers: Sgc8 for targeting the cell-surface PTK-7 protein to mediate endocytosis, and Cytochrome C aptamer (Cyt C apt) for targeting mitochondria (Fig. 1c). During the RCA reaction, the primer strand pairs with the template strand can stretch in the presence of phi29 DNA polymerase, result in the formation of a nanoflower structure as it wraps upon the magnesium pyrophosphate generated in DNA synthesis (Fig. 1d-e, Supplementary Fig. 1a-b). LbCas12a protein and in vitro-transcribed crRNA were pre-assembled to form a ribonucleoprotein (RNP) complex and efficiently loaded on the DNF by sequence hybridization. To improve the in vivo stability of the fluorophore-quencher ssDNA reporter, a circular reporter (CLR) was designed and integrated into the nano Cas12a sensor (InCasor) probe[27]. The reporter cyclization strategy and the dense spherical structure of DNF protected the reporters from nuclease degradation, which improved the stability of InCasor in 90% serum (Supplementary Fig. 1c). With these functional motifs, the integrated nano Cas12a sensor was designed to efficiently target mitochondria for reporting mtDNA mutations.

The reaction kinetics of InCasor for detecting targeted mtDNA were then characterized using an in vitro fluorescence assay, which showed rapid initial reporter cleavage and a continued gradual fluorescence increase (Fig. 1f). Dynamic light scattering (DLS) (Supplementary Fig. 1d) and transmission electron microscopy (TEM) imaging (Fig. 1g) was performed to analyze the morphology change of InCasor before and after incubation with 50 nM target dsDNA. An obvious self-degradation of InCasor was observed, demonstrating the design principle. The limit of detection of InCasor for $dsDNA_{ND4}$ analysis was calculated to be around 1 fM, considerably lower than the other control groups (Supplementary Fig. 1e-i). The increased sensitivity of InCasor may attribute to several factors. Firstly, the CLR exhibited a more favorable response to the trans-cleavage activity of Cas12a compared to single-stranded reporter (SSR); Secondly, the integration of the nanoprobe led to an increased local concentration of CLR, facilitating proximity between Cas12a/crRNA and CLR. This proximity improved reaction kinetics, further contributing to the increase of sensitivity. We then assessed the enduring stability of InCasor in 4 °C, in view of the enzymatic activity of Cas12a. During one week, the particle size, zeta potential, and mtDNA analysis performance of InCasor were not altered substantially (Supplementary Fig. 1j-k).

### Characterization of the mitochondria targeting process of InCasor

To evaluate the suitability of InCasor for imaging mtDNA in live cells, we performed a systematic study of its endocytosis, lysosomal escape, and mitochondrial targeting capabilities. Confocal images showed that InCasor with both Sgc8 and Cyt C apt was effectively uptake by HepG2 cells and co-localized well with the mitochondria, as indicated by a Pearson's correlation coefficient (PCC) of 0.48 (Fig. 2a-b, Supplementary Fig. 2a-b). The intracellular distribution of the InCasor probe was then analyzed at subcellular organelle resolution using Bio-TEM imaging (Supplementary Fig. 2c-e). $InCasor_{Cyt\ C\ apt+}$ probes were located in mitochondria, whereas no obvious $InCasor_{Cyt\ C\ apt-}$ probe was found in mitochondria (Fig. 2c, Supplementary Fig. 2f-g).

To further test the ability of InCasor for delivering Cas12a RNP into mitochondria, we tried to characterize each component in InCasor probe in the isolated mitochondria. The protease protection assay was first conducted to verify the effective delivery of Cas12a protein in mitochondria using $InCasor_{Cyt\ C\ apt+}$ (Supplementary Fig. 3a-b). As shown in Fig. 2d, proteinase K completely degraded MFN2 (mitochondrial outer membrane protein) and GAPDH (cytoplasmic protein), both employed as negative controls. In contrast, mitochondrial inner membrane proteins HSP60 remained shielded by proteinase K cleavage due to the presence of the mitochondrial membrane. Notably, a distinct Cas12a protein band of approximately 140 kDa in size was clear in the $InCasor_{Cyt\ C\ apt+}$ group with or without proteinase K treatment, indicating substantial portion of the Cas12a protein is efficiently transported into mitochondria via $InCasor_{Cyt\ C\ apt+}$. Conversely, proteinase K treatment resulted in the disappearance of the Cas12a protein in $InCasor_{Cyt\ C\ apt-}$ group, suggesting the mitochondria-targeted delivery capabilities of Cyt C aptamer. The presence of crRNA in mitochondria was further confirmed through the RCA- and LwaCas13a-based analytical methods (Supplementary Fig. 3c-h).

We then performed confocal imaging to analyze the location of separately labeled Cas12a protein and crRNA in InCasor in live cells. As shown in Fig. 2e, the two fluorophores (purple for Cas12a and green for crRNA) in $InCasor_{Cyt\ C\ apt+}$ group highly overlap with the red fluorescent signal of mitochondria. PCC analysis showed that by $InCasor_{Cyt\ C\ apt+}$ based delivery, the colocalization coefficient of Cas12a/crRNA and mitochondria increased from ~-0.3 to ~ 0.5 (Supplementary Fig. 3i). The fraction of double-positive mitochondria within mitochondrial extracts was then quantified. The results of flow cytometry showed the percentage of mitochondria displaying both FITC and Cy5 signals was 32.2% high compared to the control (Supplementary Fig. 3j), suggesting that the cytochrome C aptamer in InCasor contributes substantially to the mitochondria-targeted delivery of Cas12a/crRNA probe. In addition, a DNase protection assay was conducted to analyze the sequences of DNF inside mitochondria. After DNase treatment, we performed PCR with DNA extracted from mitochondria using DNF-specific primers, the diffuse band encircled by the red box indicated that the DNF in the InCasor probe was present inside the mitochondria (Supplementary Fig. 3k-l). Together, these results demonstrated that InCasor probe can delivery Cas12a protein and crRNA into mitochondria in living HepG2 cells.

We then investigated the influence of InCasor on mitochondrial homeostasis, using JC-1 probe and Calcein AM to test the changes of membrane potential and permeability of mitochondria, respectively. As shown in Fig. 2f, when we treated cells with $InCasor_{Cyt\ C\ apt+}$, the red fluorescence decreased and the green fluorescence increases, indicating JC-1 changed from a polymer to a monomer, and the mitochondrial membrane potential becomes lower. The decrease of green fluorescence intensity of Calcein AM in mitochondria indicates that mitochondrial permeability was significantly increased in the $InCasor_{Cyt\ C\ apt+}$ group (Fig. 2g), which consisted with a decrease in mitochondrial $\Delta\phi m$ after InCasor treatment (Fig. 2h). These results suggest that InCasor may promote the delivery of Cas12a RNP complex into mitochondria by disrupting mitochondrial homeostasis.

Another important function of InCasor is to increase $Mg^{2+}$ concentration in cells for activating the trans-cleavage activity of Cas12a. The DNF is mainly composed of DNA and magnesium pyrophosphate ($Mg_2PPi$), which can release $Mg^{2+}$ in the acid environment of lysosome[32]. ICP-MS was firstly used to characterize the intracellular $Mg^{2+}$ concentration of InCasor-treated HepG2 cells, which showed

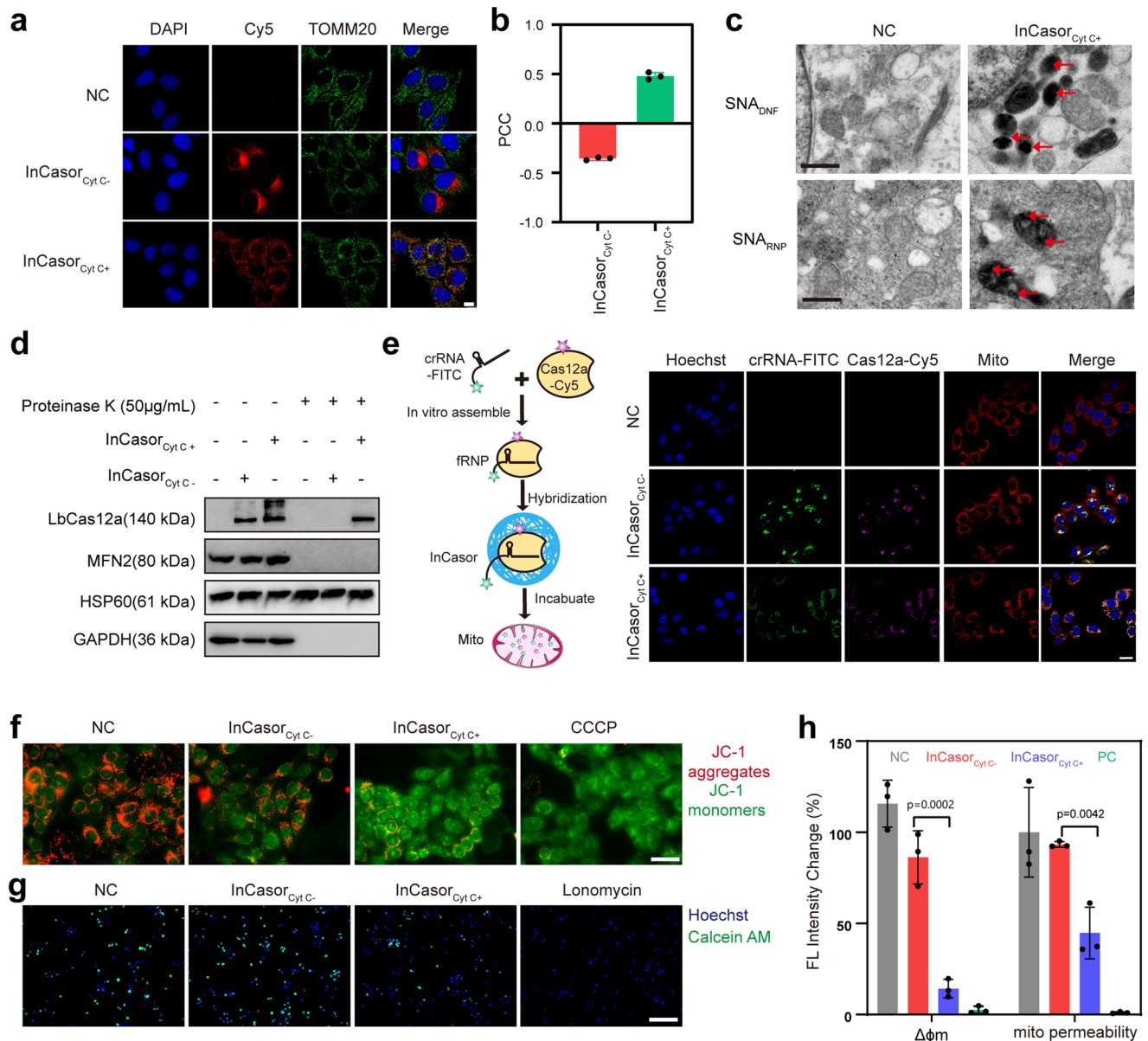

**Fig. 2 | Characterization of mitochondria targeting process of InCasor.**
**a** Confocal imaging of subcellular localization of InCasor in HepG2 cells after treated with 50 nM InCasor$_{Cyt\ C\ apt+}$ or InCasor$_{Cyt\ C\ apt-}$. InCasor were labelled with Cy5 (red), mitochondrial were stained with TOMM20 antibody (green). Scale bar: 25 μm. **b** The Pearson's correlation coefficient (PCC) of InCasor and mitochondrial were statistically analyzed by Fiji. Data are expressed as mean ± SD ($n = 3$ biologically independent experiments). **c** Bio-TEM of HepG2 cells treated with Au NPs labelled InCasor$_{Cyt\ C\ apt+}$ or InCasor$_{Cyt\ C\ apt-}$. **d** Proteinase protection assay and western blots showing successful delivery of Cas12a into mitochondria. HepG2 cells were incubated with 50 nM InCasor$_{Cyt\ C\ apt+}$ or InCasor$_{Cyt\ C\ apt-}$ for 6 h. After incubation, 20 μg of isolated crude mitochondrial fraction was treated with 50 μg/mL proteinase K for 30 min on ice, followed by western blotting. 20 μg of crude mitochondrial fraction w/o proteinase K treatment was used as control.
**e** Schematic (left) of InCasor for mitochondria targeting in living HepG2 cells by

fluorescent ribonucleoprotein consisting of synthesized fluorescent crRNA and Cas12a. Confocal imaging (right) of subcellular localization of Cas12a RNP in HepG2 cells after treated with 50 nM InCasor$_{Cyt\ C\ apt+}$ or InCasor$_{Cyt\ C\ apt-}$. Cas12a protein was labelled with Cy5 (purple), crRNA was labelled with FITC (green), and mitochondrial were stained with Mito-Tracker (red). Scale bar: 25 μm.
**f** Fluorescence microscopy images of JC-1-labeled HepG2 cells treated with 50 nM InCasor$_{Cyt\ C\ apt+}$ or InCasor$_{Cyt\ C\ apt-}$ probe at 37 °C for 6 h; scale bar: 50 μm.
**g** Fluorescence microscopy images of Calcein AM-labeled HepG2 cells treated with 50 nM InCasor$_{Cyt\ C\ apt+}$ or InCasor$_{Cyt\ C\ apt-}$ probe at 37 °C for 6 h; scale bar: 200 μm.
**h** Semiquantitative statistics of mitochondrial membrane potential (Δφm) measurements and mitochondrial permeability of HepG2 cells after different treatments. Data are analyzed by two-sided Student's t-test and shown as mean ± SD. For **a**, **c**, **d**, **e**, and **h**, the experiments were repeated three times independently. Source data from (**b**, **d**, and **h**) are provided as a Source Data file.

around 2-fold increasement than untreated control (Supplementary Fig. 4a). It's also found that, the alteration of Mg$^{2+}$ by DNA nanocarrier didn't significantly change the cell viability after 6 h treatment (Supplementary Fig. 4b). To analyze the changes of cellular Mg$^{2+}$ levels after InCasor treatment and its long-term effects on cells, Mg-Fluo-4 AM probe and ICP-MS were used to characterize the intracellular Mg$^{2+}$ levels over time. The result suggested that a large amount of Mg$^{2+}$ can

be delivered into cells by InCasor. After removing the InCasor treatment, the intracellular Mg$^{2+}$ levels would decrease over time due to the intrinsic cellular Mg$^{2+}$ regulatory mechanism, thereby maintaining intracellular Mg$^{2+}$ homeostasis (Supplementary Fig. 4c-f). Since Mg$^{2+}$ is one of the most abundant and essential divalent cations among eukaryotic cells, with an intracellular concentration range of 10-30 mM (95% is bound with ATP and other molecules, remaining 0.5-1.2 mM

unbound free)[33], $Mg^{2+}$ concentration can be regulated by various intracellular mechanism[34]. The above results suggested that InCasor was able to provide sufficient amount of $Mg^{2+}$ for activating trans-cleavage of Cas12a inside cells without significantly affect cell viability.

## Imaging of mtDNA in live cells using InCasor

To test the ability of InCasor to detect mtDNA in live cells, a crRNA targeting the *ND4* gene in mtDNA was designed to construct $InCasor_{ND4}$. In vitro analyses confirmed its mtDNA binding and self-cleavage abilities (Supplementary Fig. 5). A non-target crRNA that did not recognize any mtDNA sequence ($InCasor_{NT}$) was used as a control. In HepG2 cells treated with 50 nM $InCasor_{ND4}$ for 6 h, strong green fluorescence signals were observed, predominantly colocalized with mitochondria, while DNF/CLR and $InCasor_{NT}$ treated cells showed no obvious signals (Fig. 3a). Flow cytometry analysis further confirmed that 94.2% of HepG2 cells were lighted up by $InCasor_{ND4}$ (Fig. 3b-c).

The metabolic behavior of InCasor inside mitochondria was then examined. Specifically, a FITC labelled DNF was used to prepare FITC-$InCasor_{ND4}$ and FITC-$InCasor_{NT}$. As shown in Supplementary Fig. 6, both FITC and Cy3 signal could be observed in the mitochondria of HepG2 cells in FITC-$InCasor_{ND4}$ group after 6 h incubation, whereas only FITC signal were observed in FITC-$InCasor_{NT}$ group. The calculated half-life of FITC-$InCasor_{ND4}$ was ~9.25 h, whereas FITC-$InCasor_{NT}$ had a calculated half-life of ~11.53 h. And the half-life of Cy5 produced in response was calculated to be ~11.72 h. The shorter half-life of $InCasor_{ND4}$ may be attributed to the activation of Cas12a's trans-cleavage activity to cleave DNF, thus accelerating its self-degradation process.

qPCR and Western blot assays demonstrated that $InCasor_{ND4}$ cut mtDNA in HepG2 cells and silenced *ND4* gene expression (Supplementary Fig. 7a-b). CCK8 experimental and cell cloning experiment results showed that $InCasor_{ND4}$ probe slightly affected cell viability, possibly due to the mitochondrial homeostasis disruption and *ND4* gene silencing (Supplementary Fig. 7c-d). Whole-genome sequencing (WGS) was used to further examine the effects of $InCsaor_{ND4}$ on the mitochondrial and nuclear genomes, and no functional off-target gene editing was found in $InCasor_{ND4}$ treated cells (Supplementary Fig. 8). Then, InCasor systems with or without the designed CLR or Cyt C apt were tested to demonstrate the necessity of different functional motifs (Supplementary Fig. 9a-c). The optimal applied concentration (Supplementary Fig. 9d) and general applicability of InCasor in different cancer cells (HepG2, MDA-MB-231, MCF-7, and A549 cell lines, Supplementary Fig. 9e) were also studied. Together, these results demonstrated that $InCasor_{ND4}$ was able to efficiently recognize target mtDNA in various kinds of live cells and generate fluorescence signals for mtDNA imaging.

To demonstrate the importance of providing additional $Mg^{2+}$ in activating the trans-cleavage of Cas12a/crRNA in live cells, we prepared another $InCasor_{ND4}$-$Co^{2+}$ probe by replacing the encapsulated $Mg^{2+}$ with $Co^{2+}$ (Supplementary Fig. 10). Although both probes showed efficient mitochondrial targeting and mtDNA gene silencing ability (Fig. 3d), the $InCasor_{ND4}$-$Co^{2+}$ probe did not produce obvious fluorescence signals in live HepG2 cells (Fig. 3e-f), indicating that a sufficient $Mg^{2+}$ supply is crucial for activation of the trans-cleavage activity of Cas12a/crRNA in live cells. To further examine the necessity of providing additional $Mg^{2+}$, the cells treated with $InCasor_{ND4}$-$Co^{2+}$ probe were incubated with DPBS containing different concentrations of $Mg^{2+}$. The intracellular $Mg^{2+}$ levels and the mtDNA sensing ability of $InCasor_{ND4}$-$Co^{2+}$ were then examined using CLSM. As shown in the results, the intensity of Mg-Fluo-4 AM and the intensity of Cy3 in HepG2 cells both increased with the concentration of $Mg^{2+}$ in DPBS (Fig. 3g-j). The correlation analysis of Cy3 intensity and Mg-Fluo-4 AM showed that the sensing ability of $InCasor_{ND4}$-$Co^{2+}$ was positively correlated with the intracellular $Mg^{2+}$ level (Fig. 3k). Collectively, these data suggest that increasing intracellular $Mg^{2+}$

levels contributed to the intracellular triggering the trans-cleavage activity of Cas12a.

## InCasor enables the detection of mutated mtDNA in live cells

Eukaryotic cells contain numerous copies of circular mtDNA, which can coexist as a heteroplasmic mixture comprising both wild-type and mutant alleles[35]. Discriminating mtDNA SNVs in live cells may facilitate mtDNA mutant identification and heterogeneity analysis[11]. As Cas12a holds the potential for identifying SNV[30,36], we were interested in determining whether the InCasor system could be used to discriminate SNV in mtDNA in live cells. Since the protospacer-adjacent motif (PAM) sequence is necessary for Cas12a to bind to dsDNA[20], which is essential for activating trans-ssDNA cutting, we first hypothesized that InCasor could be used to identify mitochondria with specific mtDNA mutations in live cells using PAM recognition mechanism.

To test this idea, we sequenced the mtDNA genome in 2 human carcinoma cell lines (HepG2 and MDA-MB-231), identified an SNV in the *ND4* gene (12084 C > T) in MDA-MB-231 cells (Fig. 4a) and used the mutant sequence as a PAM to design a Cas12a/crRNA probe. Using the designed crRNA, an InCasor probe targeting the 12084 C > T mutation in the *ND4* gene ($InCasor_{ND4-2-MT}$) was constructed and tested using in vitro cleavage assay and single-cell imaging analysis (Fig. 4b; Supplementary Fig. 11a-b). Flow cytometry (Fig. 4c) and confocal imaging of live cell co-culture experiments (Fig. 4d-e) both demonstrated the ability of $InCasor_{ND4-2-MT}$ to identify the 12084 C > T mutation in MDA-MB-231 cells, as the sensor produced a green signal predominantly in the cytoplasm (appearing yellow when colocalized with DiD). To exclude the effect of cellular environment, we extracted mitochondria with different genotypes from wild-type and mutant cell lines, and imaged the fluorescent signal produced by $InCasor_{ND4-2-MT}$. As shown in Fig. 4f, the Cy3 fluorescence signal generated by $InCasor_{ND4-2-MT}$ probe only co-localized with the green fluorescence-labelled MT-Mito, which can effectively distinguish mitochondrial genotypes.

Recognition of the PAM endowed Cas12a/crRNA with high SNV specificity but limited its application to a small subset of mtDNA sequences. To expand the sequence applicability of InCasor probes, we tried to engineer crRNA for wider sequence application. Several crRNA modification methods have been reported for enhancing the SNV specificity of CRISPR-Cas12a, such as introducing additional mismatches in the spacer, designing secondary structures on the spacer, and employing chimeric crRNA with DNA components[37–40]. In this work, we designed a InCasor probe for the detection of a PAM-adjacent SNV in MDA-MB-231 cells (13105 A > G in the *ND5* gene) (Supplementary Fig. 11e). An in vitro fluorescence assay showed that for this specific target, the mutant mtDNA sequence could not be distinguished from the wild-type sequence when using unmodified fully matched crRNA ($crRNA_0$). Therefore, we engineering modified crRNAs with the addition of mismatches near the SNV site ($crRNA_{1-4}$, Fig. 4g). We found that $crRNA_1$ and $crRNA_2$, which had one and two nucleotide mismatches, respectively, showed greatly improved specificity without significantly affecting the detection sensitivity (Fig. 4g, Supplementary Fig. 11f). To achieve higher sensitivity, we used $crRNA_1$ to construct $InCasor_{ND5-MT-1}$ for imaging of the 13105 A > G mtDNA mutation in MDA-MB-231 cells. $crRNA_0$ was used to construct $InCasor_{ND5-MT-0}$ as a control. $InCasor_{ND5-MT-1}$ with modified $crRNA_1$ was able to distinguish MT-Mito from WT-Mito, while the unmodified $InCasor_{ND5-MT-0}$ didn't show the specificity (as indicated by the purple signals for both MT-Mito and WT-Mito) (Fig. 4h-I, Supplementary Fig. 11g). Isolated mitochondria imaging results (Supplementary Fig. 11h) also suggested the improved specificity of modified crRNA.

To further examine InCasor's ability to recognize SNVs distant from the PAM sequence, we selected two loci with mutations at position 12 or 16 nt after PAM according to sequencing results or literature[41,42], respectively, and modified the crRNA to test its recognition by InCasor. Compared to fully matched crRNA and $crRNA_{SM1}$,

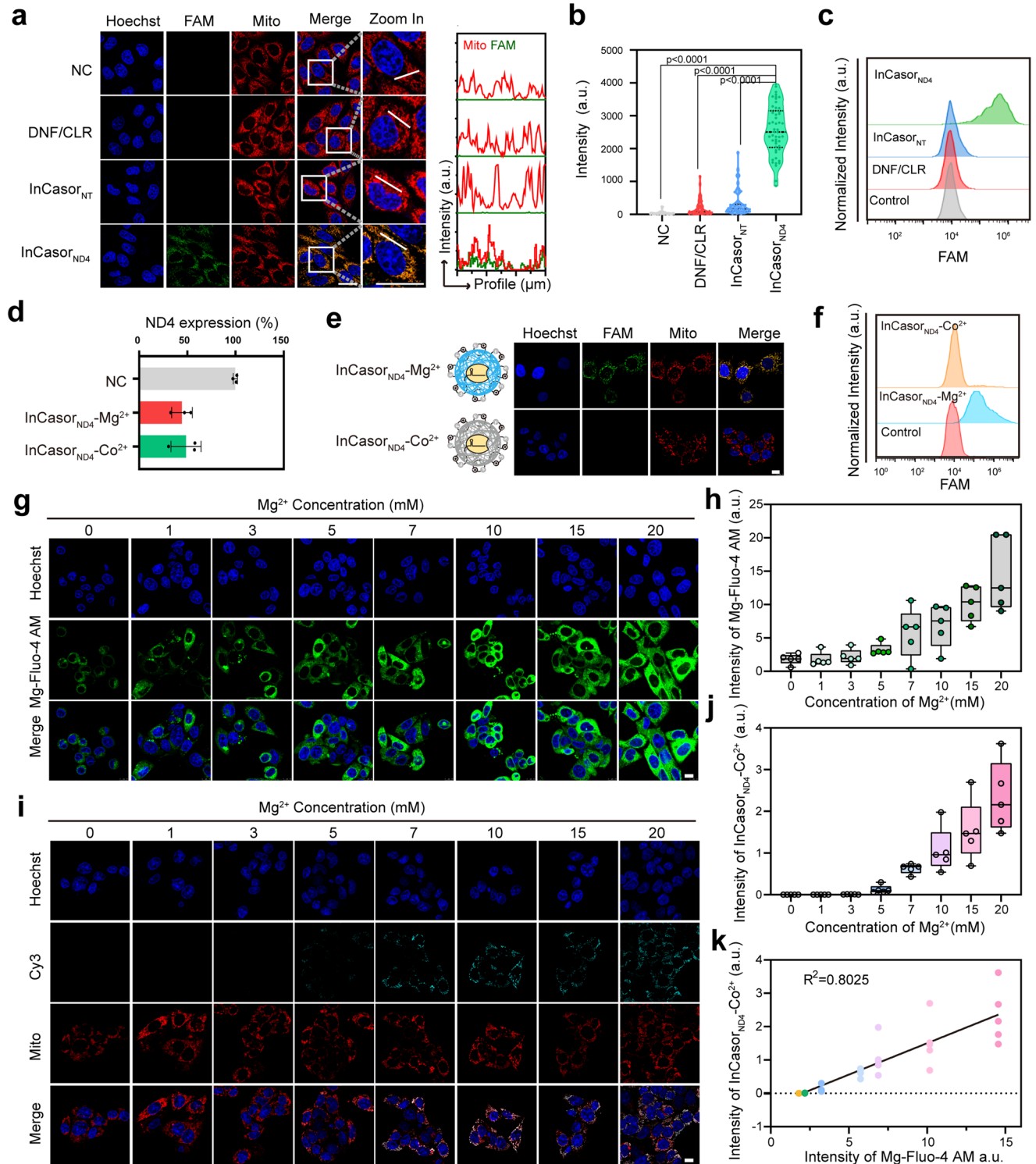

**Fig. 3 | InCasor_ND4 for imaging *ND4* gene in mtDNA in live cells. a** Confocal imaging of HepG2 cells treated with DNF/CLR, InCasor_NT and InCasor_ND4. Scale bar: 10 μm. The spatial co-localization of InCasor and mito were statically analyzed by Fiji. **b** Semi-quantitative analysis of fluorescent intensity in each cell is shown. $n = 50$ biologically independent cells. $P < 0.0001$ determined by via one-way ANOVA with Tukey's post-test. **c** Flow cytometry analysis of groups corresponding to (**b**). **d** Analysis of *ND4* expression in HepG2 cells treated with InCasor_ND4-Mg²⁺ or InCasor_ND4-Co²⁺ using qPCR. n = 3 biologically independent experiments, data show mean ± SD. **e, f** Confocal imaging (**e**) and flow cytometry analysis (**f**) of HepG2 cells treated with InCasor_ND4-Mg²⁺ or InCasor_ND4-Co²⁺. Scale bar: 10 μm. The experiments were repeated three times independently. **g** Confocal imaging was used to analyze the fluorescence signal of Mg-Fluo-4 AM in HepG2 cells cultured in DPBS with different concentrations of Mg²⁺ for 12 h. Scale bar: 10 μm. **h** Semi-quantitative statistics of Mg-Fluo-4 AM in HepG2 cells after different treatments. $n = 5$ biologically independent experiments. **i** Confocal imaging of HepG2 cells treated with InCasor_ND4-Co²⁺ in DPBS containing different concentrations of Mg²⁺. Scale bar: 10 μm. **j** Semi-quantitative analysis of fluorescent intensity in panel i is shown. **h, j** Box-and-whisker plots: the boxes extend from the first to the third quartile, the middle lines denote the median and the whiskers indicate the minimum and the maximum value range. $n = 5$ biologically independent experiments. **k** Correlation analysis of the fluorescence intensity of InCasor_ND4-Co²⁺ and the fluorescence intensity of Mg-Fluo-4 AM. $n = 5$ biologically independent experiments, data show mean ± SD. Source data from (a, b, d, h, j, and k) are provided as a Source Data file.

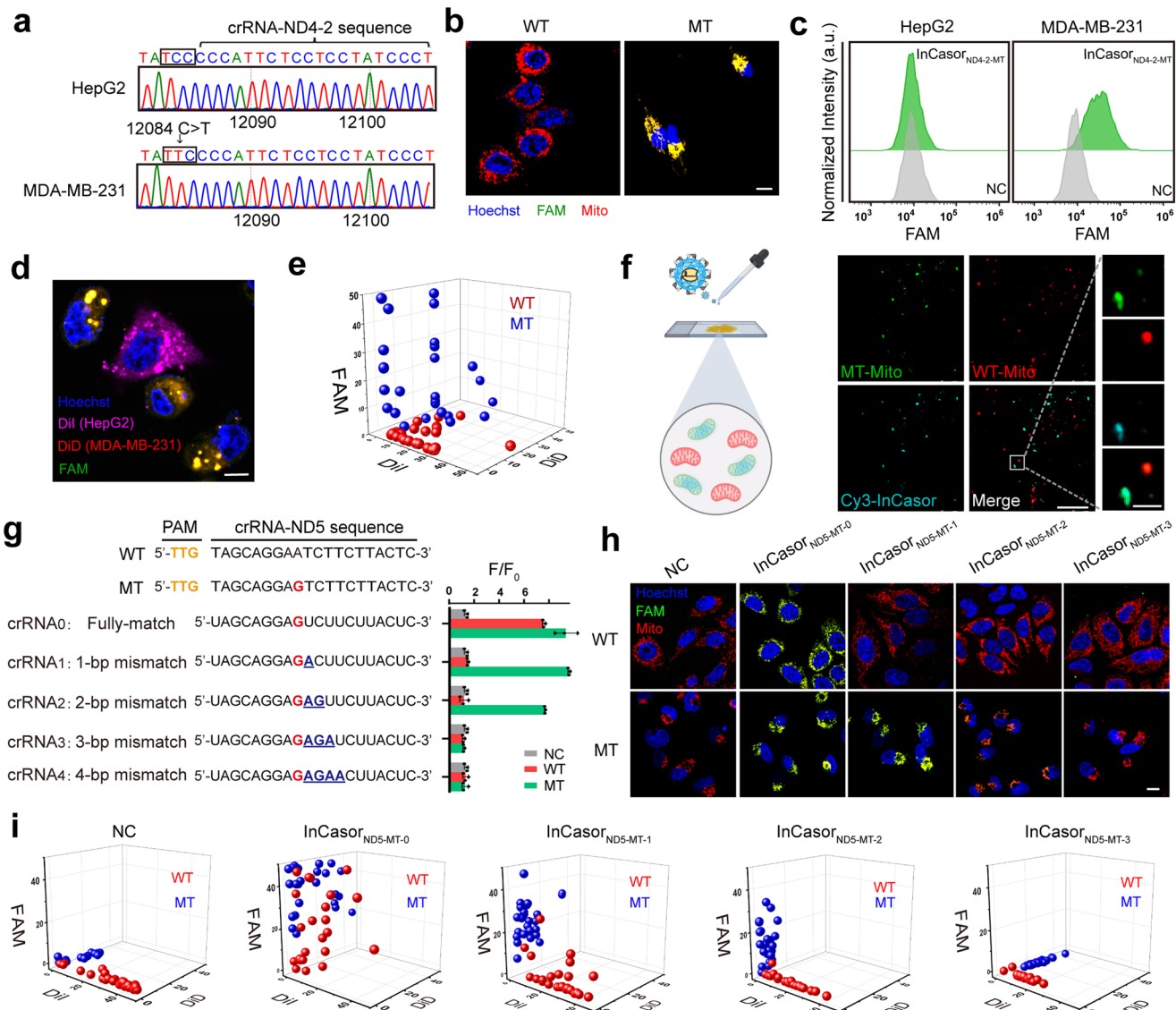

**Fig. 4 | InCasor for imaging of mtDNA mutations in live cells. a** Sequencing identification of an SNV (12084 C > T) in mtDNA (*ND4* gene) in MDA-MB-231 cells (MT). HepG2 cells were used as the wild-type control (WT). **b, c** InCasor$_{ND4-2-MT}$ was capable of sensitive detection of the 12084 C > T mtDNA mutation in live cells. Confocal imaging (**b**) and flow cytometry (**c**) were used to analyze MDA-MB-231 and HepG2 cells incubated with InCasor$_{ND4-2-MT}$. The experiments were repeated three times independently. Scale bar: 10 μm. **d, e** Confocal imaging were used to analyze MDA-MB-231 and HepG2 cells incubated with InCasor$_{ND4-2-MT}$. DiI stained HepG2 cells, purple; DiD stained MDA-MB-231 cells, red; Hoechst: nucleus, blue; FAM: InCasor$_{ND4-2-MT}$, green. Semi-quantitative statistics are performed on the three types of fluorescence intensities (DiI, DiD and FAM) of each cell in the co-culture group. (n = 50). Scale bar: 10 μm. **f** Confocal imaging of mitochondria extracted from MDA-MB-231 cells and HepG2 cells using InCasor$_{ND4-2-MT}$ to detect mutant

mtDNA $_($12084 C > T). Scale bar: 25 μm. The experiments were repeated three times independently. **g** Sequences of crRNA$_0$ (fully matched with the 13105 A > G mutant *ND5* gene) and crRNA$_1$-crRNA$_4$ (with different numbers of nucleotide mismatches). Each SNV is indicated by a bold red letter. Each mismatched position is indicated by an underlined bold blue letter. An in vitro fluorescence assay was used to estimate the ability of different InCasor probes to detect the 13105 A > G mutation in the *ND5* gene. *n* = 3 biologically independent experiments, data show mean ± SD. The sequences of all crRNAs used are listed in Supplementary Table 1. **h** Confocal imaging of HepG2 cells (WT) and MDA-MB-231 cells (MT) treated with InCasor probe with different crRNA. Scale bar: 10 μm. **i** Semi-quantitative statistics of three fluorescence intensities (DiI, DiD and FAM) in co-cultured cells under different treatments. (*n* = 50 biologically independent cells). Source data from (c, e, g, and i) are provided as a Source Data file.

the insertion of two discrete mismatches in the seed region (crRNA$_{DM1}$) and the insertion of 1 mismatch in the SNP locus (crRNA$_{SM2}$) both showed similarly improved discrimination factors (DFs) (for mt.4769 A > G, DF of crRNA$_{SM2}$ was 4.1, DF of crRNA$_{DM1}$ was 3.6; for mt.3916 G > A, DF of crRNA$_{SM2}$ was 2.4, DF of crRNA $_{DM1}$ was 2.2) (Supplementary Fig. 12). Overall, engineering the crRNA contributed to InCasor's ability to sense SNV sites distant from the PAM site.

The discrete signals generated by InCasor in individual mito-chondria suggested the potential usefulness of this sensor for studying the distribution of mutant mitochondria in heteroplasmic cells. To test this potential, we first created cells with heteroplasmic mtDNA, i.e.,

containing both mitochondria with wild-type mtDNA (WT-Mito) and mitochondria with mutant mtDNA (MT-Mito), by isolating mitochon-dria from MDA-MB-231 cells and transfecting them into HepG2 cells. The genotype of mitochondria in HepG2 and MDA-MB-231 cells were firstly examined by pyrosequencing (Fig. 5a). Then, the isolated MT-Mito were pre-stained with CellMask (green) and the WT-Mito in host cells were labeled with MitoTracker (red). The monitoring of mito-chondria transfection process and the heterogeneity kinetics of the heterogeneous cells (Supplementary Fig. 13a-d) indicated successful construction of heteroplasmic cell model. Furthermore, in situ RCA[8] was used to detect the genotype of heterogeneous cells in situ, with

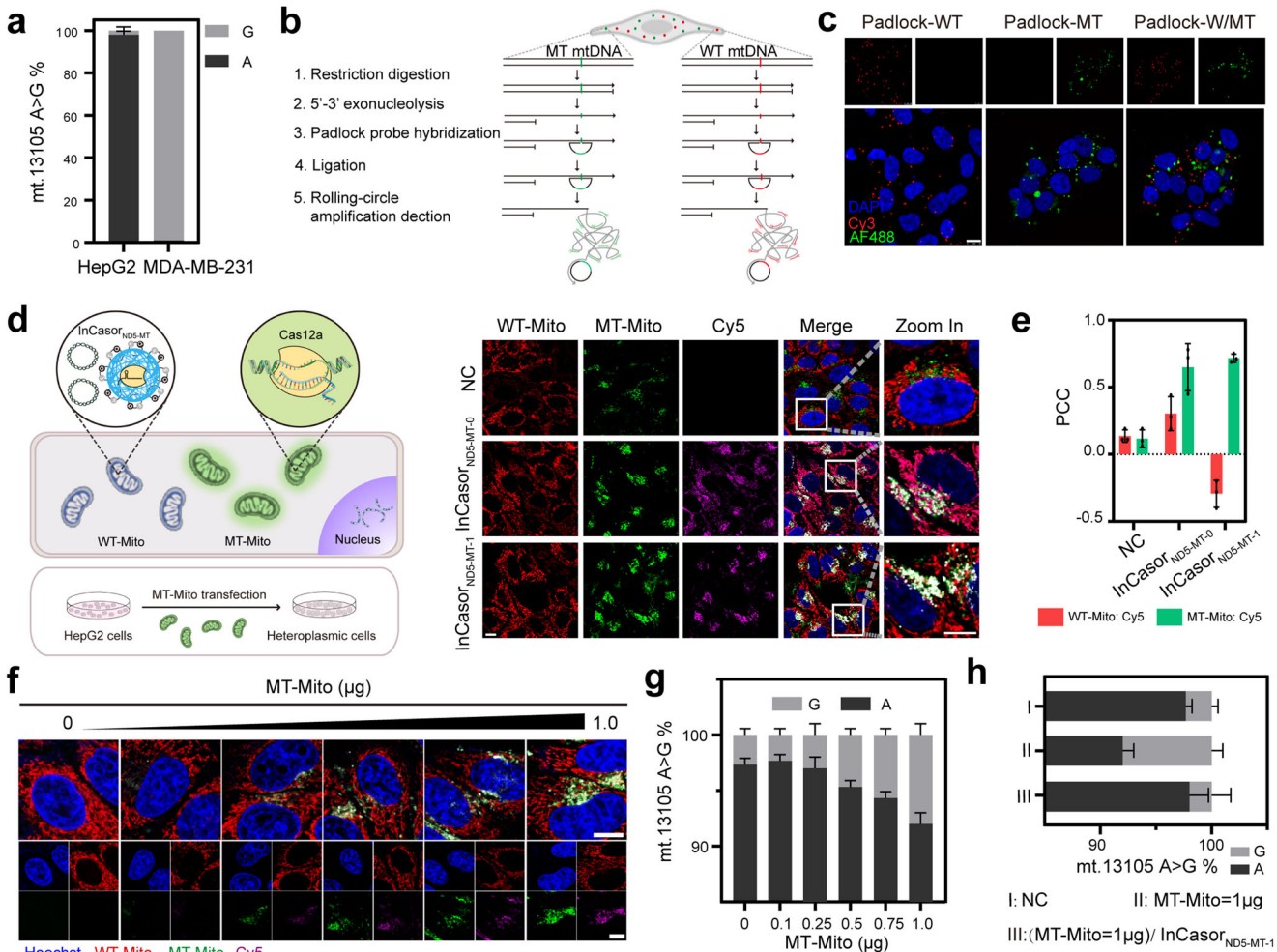

**Fig. 5 | InCasor for imaging of mtDNA mutations in live heteroplasmic cells. a** Pyrosequencing analysis of HepG2 cells and MDA-MB-231 cells at mt. 13105 A. (n = 3). Data are presented as mean ± SD. **b, c** In situ genotyping of the heterogeneous cells using rolling circle amplification method. Nuclei were labelled by DAPI (blue), Cy3-labeled probe for imaging wild-type mitochondrial DNA (red) and AF488-labeled probe for imaging mutant mitochondrial DNA (green). Scale bar: 10 μm. The experiments were repeated three times independently. **d** Analysis of InCasor$_{ND5-MT-0}$ and InCasor$_{ND5-MT-1}$ for imaging mtDNA mutation (13105 A > G) in live heteroplasmic cells. Scale bar: 10 μm. **e** PCC of purple signal with green or red fluorescence in panel d were statically analyzed. $n$ = 3 biologically independent experiments, data show mean ± SD. **f** Representative images of HepG2 cells

transfected with different amounts of MT-Mito (0, 0.1, 0.25, 0.5, 0.75, and 1.0 μg) and then treated with InCasor$_{ND5-MT-1}$. Cell nuclei are stained blue by Hoechst 33342, WT-Mito were labeled with MitoTracker (red), and MT-Mito were pre-stained with CellMask (green); the fluorescence generated by InCasor$_{ND5-MT-1}$ (Cy5) appears purple. Scale bar: 10 μm. **g** Quantification of heteroplasmy after transfection different amounts of MT-Mito into HepG2 cells (0, 0.1, 0.25, 0.5, 0.75, and 1.0 μg) (mean ± SD; $n$ = 3 biologically independent experiments). **h** Quantification of heteroplasmy of HepG2 cells transfected with 1.0 μg MT-Mito for 12 h and then treated with InCasor$_{ND5-MT-1}$ for 6 h (mean ± SD; $n$ = 3 biologically independent experiments). Source data from (a, c, e, g, and h) are provided as a Source Data file.

distinct, robust signals showing the distribution of the two variants of mitochondrial genome in the heterogeneous cells (Fig. 5b-c, Supplementary Fig. 13e-f).

To compare the SNV imaging specificity between InCasor$_{ND5-MT-0}$ and InCasor$_{ND5-MT-1}$, we used InCasor$_{ND5-MT-1}$ for imaging of the 13105 A > G mtDNA mutation in the constructed heteroplasmic cells and used InCasor$_{ND5-MT-0}$ as a control. Confocal fluorescence imaging data indicated that InCasor$_{ND5-MT-1}$ with modified crRNA$_1$ was able to distinguish MT-Mito from WT-Mito in a heteroplasmic cell, while InCasor$_{ND5-MT-0}$ was not (as indicated by the purple signals for both MT-Mito and WT-Mito) (Fig. 5d-e, Supplementary Fig. 13g-h). In addition, we quantified the intensities of three fluorescence channels for 50 single mitochondria in heterogeneous cells with different probe treatment groups (Supplementary Fig. 13i). Compared with InCasor$_{ND5-MT-0}$ group, the FAM fluorescence intensity of WT-Mito in InCasor$_{ND5-MT-1}$ group was significantly reduced while the distribution of MT-Mito in the 3D map did not change significantly, suggesting that

a more general sequence applicability of InCasor can be achieved through crRNA modification.

To demonstrate InCasor is able to quantitively detect mitochondria containing mutant DNA in living cells, we tried to transfect different amount of MT-Mito into HepG2 cells and analyze them with InCasor. As shown in Fig. 5f, when the cells were incubated with larger amount of MT-Mito, more green fluorescent signals appeared in the cytoplasm of HepG2 cells. When treating the cells with InCasor$_{ND5-MT-1}$, a corresponding Cy5 fluorescence signal (purple) was generated, which were colocalized well with the MT-Mito signal (Fig. 5f and Supplementary Fig. 14a). Statistical analysis showed synchronous increases in the green and purple signals with high colocalization (Supplementary Fig. 14b-c). Pyrosequencing was also used to confirm the altered heterogeneity value in cells treated with different amount of MT-Mito (Fig. 5g). Interestingly, we found that, with InCasor$_{ND5-MT-1}$ treatment, the mutation content of mtDNA was reduced from 8.0 ± 1.0% to 2.3 ± 0.6% (Fig. 5h), indicating that InCasor$_{ND5-MT-1}$ was

able to edit mutated mtDNA and reduce the mutation content in heteroplasmic cells.

### In vivo imaging of mutated mtDNA

In vivo imaging of SNV mutations in mtDNA has been a long-sought goal, which has not been achieved before. One major advantage of InCasor is its integrated nanostructure, which holds potential for in vivo imaging of tissues harboring mtDNA mutations. To test this application, we first analyzed the in vivo biodistribution of InCasor after tail vain injection into HepG2-tumor-bearing mice. Benefiting from the nanometer size, the circulating half-life of InCasor in the second phase ($t1/2(\beta)$) was calculated to be 2.63 h, which was significantly higher than that of ssDNA (1.33 h) (Supplementary Fig. 15a). With Sgc8 aptamer, InCasor was able to specifically target tumor tissue in live mice (Supplementary Fig. 15b-e). After circulation, most of InCasor would be expelled from the circulatory system through kidney (Supplementary Fig. 15f).

In vivo fluorescence imaging and ex vivo organ fluorescence imaging were also conducted to assess the in vivo stability of InCasor (Supplementary Fig. 16). Due to the integrated nanostructure design, InCasor showed remarkable stability during blood circulation, which reduces the occurrence of false positive signals and lays the foundation for its accurate sensing of SNVs in mtDNA in vivo. Hematoxylin–eosin (HE) and TUNEL staining of tumor tissues and major organs, together with hematology demonstrated the low toxicity of InCasor for in vivo application (Supplementary Fig. 17).

To assess the ability for in vivo imaging of mtDNA mutations of InCasor, two cell lines derived from different types of tumors (HepG2 and MDA-MB-231) were used to construct tumor-bearing mouse models, since their mtDNA sequences were confirmed and the corresponding InCasor probes targeting different mtDNA mutation (*ND4* and *ND5*) have been designed and tested in live cells (Fig. 4). Mouse models bearing either a HepG2 (WT) or MDA-MB-231 (MT) tumor were firstly intratumorally injected with InCasor$_{ND4}$ or InCasor$_{ND4-2-MT}$, which were used to detect the wild type *ND4* gene or 12084 C > T mutation (Fig. 6a). As shown in Fig. 6b-c, InCasor$_{ND4}$ produced obvious fluorescence signals in both HepG2 and MDA-MB-231 tumor tissue, while the probe targeting the mutant *ND4* gene (InCasor$_{ND4-2-MT}$) only produced signals in MDA-MB-231 tumors, demonstrating the ability of InCasor for in vivo imaging of an mtDNA mutation. To confirm these results, other animal models were also assessed, including dual-tumor models treated with intratumorally injection of InCasor$_{ND4}$ and InCasor$_{ND4-2-MT}$ or tail vein injection of InCasor$_{ND4-2-MT}$ (Supplementary Fig. 18).

To further demonstrate the in vivo SNV imaging ability of InCasor, an experimental scheme by tail vein injection of InCasor$_{ND5-MT-0}$ or InCasor$_{ND5-MT-1}$ into dual-tumor mice (bearing both HepG2 and MDA-MB-231 tumors) was performed (Fig. 6d). Eight hours after tail vein injection of InCasor$_{ND5-MT-0}$, obvious fluorescence signals were observed not only in the HepG2 and MDA-MB-231 tumors but also in the whole mouse body (Fig. 6e-f). This finding illustrated the low SNV specificity of InCasor$_{ND5-MT-0}$, which was consistent with the earlier live cell imaging results. In contrast, InCasor$_{ND5-MT-1}$ produced obvious fluorescence signals only in MDA-MB-231 tumors, demonstrating the capability of InCaso$_{ND5-MT-1}$ for imaging the 13105 A > G mtDNA mutation in a mouse model. Other animal models, including those treated by intratumorally injection of InCasor$_{ND5-MT-0}$ or InCasor$_{ND5-MT-1}$ into different tumors, are shown in Supplementary Fig. 19. The results obtained with these models suggested that InCasor holds the potential for efficiently identifying SNVs in mtDNA in vivo.

In vivo fluorescent imaging can only provide tissue-level resolution. To examine the actual spatial resolution of InCasor for imaging mtDNA mutations in tissue, we sacrificed the mice used in the Fig. 6e and did tissue slices imaging to test whether InCasor is able to identify metastatic tumor cells. Interestingly, we found obvious white nodules

on the liver surface, indicating tumor metastasis in the liver. Ex vivo imaging of dissected liver tissues presented obvious fluorescence signal (Fig. 6g). The liver tissue was then sectioned and subjected to HE staining and immunofluorescence staining for Arginase-1 (green, normal liver cells) with Cy5 fluorescence representing InCasor (red). HE staining showed typical tumor metastases (small cube-shaped cells, fatty degeneration and clear boundaries) in the liver tissue (Fig. 6h). A clear separation of red and green signal was observed in 2 different liver tissue sections by immunohistochemical staining (Fig. 6h). The PCC calculated for the colocalization between the green and red signals was approximately -0.4 (Supplementary Fig. 19d), indicating obvious separation. These findings indicated that InCasor may be useful for imaging of tumor metastases by visualizing cells with mutant mtDNAs in vivo.

## Discussion

Mutations in mtDNA underlie a substantial portion of the mitochondrial disease burden. Massively parallel sequencing has uncovered extreme genetic variation within mtDNA[43], which can be utilized to identify mitochondrial abnormalities associated with genetic disorders[44,45]. However, the in vivo distribution of mtDNA mutations is less known[1]. Although some methods have been developed for imaging of mtDNA mutations in fixed cells and tissue sections[8,11], strategies for in vivo imaging of SNVs in mtDNA have not been reported.

Here, we developed an integrated nano Cas12a sensor (InCasor) for imaging of mtDNA mutations in vivo. Our results showed that InCasor can robustly sense and report specific mtDNA sequence in living cells. By engineering crRNA, InCasor could be used to distinguish SNVs, enabling precise and sensitive detection of mitochondria with mutant sites in co-culture cells and mtDNA heterogeneous cells. These properties further led to its application in identifying tumor tissue and metastasis in vivo.

A key highlight of this study is the demonstration of mitochondria-targeted delivery of the Cas12a/crRNA complex via InCasor, which is considered a major challenge. Due to the bilayer membrane structure of mitochondria, transporting guide RNA and Cas proteins into mitochondria is generally considered to be highly difficult[46]. To address this issue, this study introduced nanoparticle-based mitochondria-targeted delivery strategy, which has been used for successfully delivery of proteins or nucleic acids into mitochondria for many therapeutic or diagnostic applications[47–49] by incorporating mitochondrial targeting motifs, such as triphenylphosphine[50,51]. In this work, we integrated Cas12a/crRNA complexes into a nucleic acid-based nanocarriers with Cyt C aptamers for targeted delivery of Cas12a/crRNA into mitochondria. Our results showed that InCasor can effectively deliver Cas12a/crRNA complexes into mitochondria by affecting the permeability of the mitochondrial membrane.

Due to the limited copy number of mutant mtDNA in cells, a high signal amplification strategy is needed. In this work, using $Mg^{2+}$ to enhance the trans-cleavage activity of Cas12a in live eukaryotic cells, we can efficiently report mtDNA mutations in live cells and in vivo. The integrated design of InCasor further leads to high local concentrations of Cas12a/crRNA and reporters inside mitochondria, providing high sensitivity and signal-to-noise ratio for mtDNA analysis. The use of nanocarriers and cyclized reporter also greatly enhanced the probe stability in vivo, reducing the false-positive signals. Moreover, the crRNA engineering process substantially enhanced Cas12a's ability for recognition of SNVs in non-PAM sequences.

Although promising, InCasor is currently difficult to be used for analyzing nuclear genomic mutations in living cells. Besides, InCasor potentially cuts to destroy the target DNA sequence while detecting, which inevitably affect the cell viability during the sensing process. The potential solution may be finding some new Cas effectors or engineering a new Cas12a protein, which can recognize target mtDNA and induce trans-cleavage, but won't cut target sequence. The use of

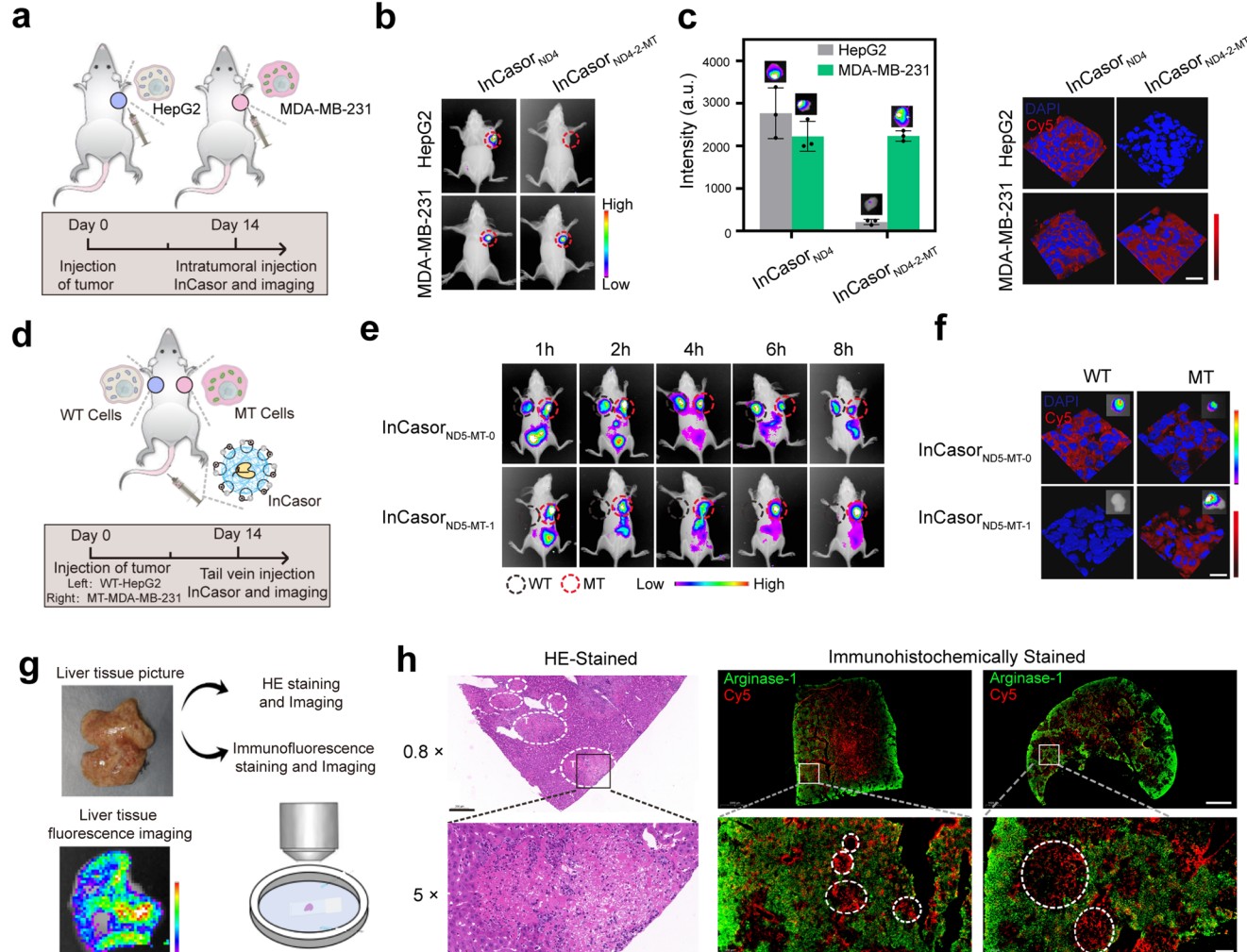

**Fig. 6 | In vivo imaging of mtDNA mutations in a mouse model using InCasor.**
**a** Schematic of intratumoral injection of InCasor_ND4 or InCasor_ND4-2-MT into a mouse model bearing either a HepG2 (WT) or MDA-MB-231 (MT) tumor to detect 12084 C > T mtDNA mutations in vivo. **b** In vivo imaging of different tumor-bearing mice after injection of InCasor_ND4 or InCasor_ND4-2-MT. The experiment was independently repeated three times with similar results. **c** Evaluation of the fluorescence intensity of dissected mouse tumor tissue. $n = 3$, data show mean ± SD. The length and width of the slices are both 2000 μm. **d** Schematic of tail vein injection of InCasor_ND5-MT-0 or InCasor_ND5-MT-1 into a dual-tumor mouse model (bearing a HepG2 tumor (WT) on the left and an MDA-MB-231 tumor (MT) on the right) to detect the 13105 A > G mtDNA mutation in vivo. **e** Time course analysis of fluorescence intensity in the dual-tumor mouse model illustrated in (**d**). Fluorescence signals were detected only in MT tumors over 8 h after tail vein injection of InCasor_ND5-MT-1,

indicating improved specificity compared with that of InCasor_ND5-MT-0 (which produced fluorescence signals in both tumors). $n = 3$ biologically independent samples. **f** Representative images of the tumor tissue and tumor tissue sections dissected from (**e**). Scale bar: 25 μm. **g** Images of dissected liver tissue from mice 8 h after intravenous injection of InCasor showing obvious white nodules and fluorescence signals. **h** Hematoxylin and eosin (HE) and immunohistochemical staining of liver tissue sections from the liver in (**g**). Immunohistochemically stained liver tissue sections are shown on the right. The green, red, and blue colors correspond to Arginase-1, the Cy5 reporter from InCasor_ND5-MT-1 and DAPI, respectively. The white dotted lines separate the tumor lesions (T) from normal tissue. The scale bars correspond to 2000 μm (0.8× images) and 200 μm (5× images). Source data from c are provided as a Source Data file.

fluorochromes also limit InCasor's applicability in in vivo imaging. To increase the versatility of InCasor's imaging, some other signal output modalities, such as MRI and PET scanning modalities[52,53], may be further investigated.

As long as we know, InCasor is a powerful tool to enable direct visualization of cells with mtDNA mutations in vivo, fulfilling a long-sought goal[4]. The monitoring of mtDNA mutations will allow the estimation of mtDNA heterogeneity and aid in the identification of tumor tissue with specific mtDNA mutations. Future work would focus on continued development of InCasor for nuclear genome analysis and high spatiotemporal resolution tracking of SNV sites to fully demonstrate the usefulness of InCasor for various in vivo gene mutation-related basic research, diagnostic, and gene therapy applications.

## Methods

### Ethical statement

All microimaging experiments were approved by the Modern Analysis and Computing Center of Zhengzhou University. The protocol for animal experiment was performed according to the guidelines of the regional Animal Experimentation Ethics Committee and Zhengzhou University. The animal laboratory holds accreditation number 110322211102955054. All experiments followed all relevant guidelines and regulations.

### Preparation of nucleic acid

Oligonucleotides were synthesized commercially by Sangon Biotech, China. To perform the fluorophore-quencher ssDNA cleavage assay, double-stranded DNA were annealed in 1× hybridization buffer (in a 1:1

molar ratio of target (TS) and non-target (NTS) strands). The annealing process involved heating at 95℃ for 5 minutes, followed by slow cooling in the Thermal Cycler (BIO-RAD).

crRNAs were synthesized using in vitro transcription with HiScribe™ T7 Quick High Yield RNA Synthesis Kit from NEB. The DNA templates contains T7 promoter and full-length crRNAs. The synthesized crRNAs were subsequently purified using TRIzon Reagent (CWBIO, China). We extended the original Cas12a crRNA backbone sequence at the 5′ end by 20 nt based on previously reported extension method[54,55] to load Cas12a/crRNA ribonucleoprotein (RNP) complex onto InCasor.

### Preparation of InCasor probe

**Preparation of DNA-Mg²⁺ hybrid nanoflower.** The DNA-$Mg^{2+}$ hybrid nanoflower (DNF) were prepared by rolling circle amplification. Specifically, the RCA template was formed by annealing primer 1 and primer 2 with the 5 'phosphorylated Template 1 (encoding Sgc8 aptamer and binding domain of Cas12a/crRNA complex) and Template 2 (encoding Cytochrome C aptamer (Cyt C apt) and binding site for circular reporter (CLR)) in 1× T4 DNA Ligase buffer, and then cyclized with T4 DNA ligase (Thermo Fisher Scientific). The RCA reaction was carried out in a 100 μL mixture containing 1 μM RCA template, 2.5 mM dNTP (Takara Biotechnology Co), 0.25 U/μL Phi 29 DNA polymerase (Thermo Fisher Scientific) and 1× Phi 29 DNA polymerase buffer. Incubate the reaction mixture at 37 °C for 2 hours and then denature at 65 °C for 10 minutes. After undergoing centrifugation at 12,000 g for 20 min, the precipitate was carefully collected, washed, twice with DNase-free $H_2O$ and reconstituted with 40 μL of DNase-free $H_2O$. To assess the synthesized DNFs, 1% agarose gel electrophoresis was carried out. The particle size and Zeta potential of DNF were measured with Zetasizer (Malvern). Both scanning electron microscopy and transmission electron microscopy were employed to scrutinize and analyze the morphology of DNFs.

**Preparation and characterization of DNA-Co²⁺ hybrid nanoflower.** A DNF template encoding MUC1 aptamer was designed to synthesize $DNF_{MUC1}$-$Mg^{2+}$ and $DNF_{MUC1}$-$Co^{2+}$. $DNF_{MUC1}$-$Mg^{2+}$ were synthesized by the above-described method. The preparation and characterization of $DNF$-$Co^{2+}$ hybrid nanoflowers were as follows. $Template_{MUC1}$ was cyclized using CircLigase II (Lucigen). In brief, 5 μL 10 mM 5 'phosphorylated $Template_{MUC-1}$ and 0.1 U/μL CircLigase II ssDNA Ligase in 1× CircLigase II reaction buffer containing 2.5 mM $CoCl_2$ was incubated at 60 °C for 16 h, and denatured at 80 °C for 10 minutes. A solution containing 50 μL 1 mM cyclized $Template_{MUC-1}$, 7.5 μL 10 mM $Primer_{MUC-1}$, 1× RCA reaction buffer (50 mM Tris buffer (pH 7.0), 10 mM $CoCl_2$, 10 mM $(NH_4)_2SO_4$) was heated to 95 °C and slowly cooled to room temperature (0.1 °C/s). After that, 20 μL 10 mM dNTPs and 5 μL 10,000 U/mL Phi 29 DNA polymerase were introduced into the mixture. After incubation at at 30 °C for 20 h, the samples were heated at 65 °C for 10 minutes. The resulting precipitate was washed 5 times by resuspending in DEPC-treated $H_2O$ after collected by centrifugation. Then the DNFs were analyzed by agarose gel electrophoresis, DLS and SEM-based EDS.

**Preparation of circular fluorophore-quencher reporter (CLR).** To prepare CLR, the single-stranded DNA (ssDNA) with fluorescent-quencher labelling were firstly dissolved in T4 DNA Ligase buffer in appropriate concentration, which was then heated at 95℃ for 5 minutes and slowly cooled to 25℃.The formed circular structure with a nick were then incubated with T4 DNA ligase at overnight 25℃ to form CLR. Finally, the ligase was denatured at 75℃ for 10 minutes. The CLR were extracted via phenol-chloroform extraction and ethanol precipitation. The concentration was determined by NanoDrop2000 (Thermo). The reporter sequence is listed in the Supplementary Table 1.

**Preparation and characterization of InCasor probe.** To form Cas12a/crRNA complex, LbCas12a (TOLOBIO, 32108-03) and crRNA (1: 1.5 molar ratio) were initially combined in TOLOBIO buffer 3 and incubated for 10 minutes at room temperature. DNF and CLR (1: 20 molar ratio) were mixed in PBS. Following a one-hour incubation at room temperature, any unbound CLR was eliminated by subjecting the mixture to centrifugation at 12,000 g for 20 minutes. To form InCasor, the DNF/CLR combination was subsequently incubated with Cas12a/crRNA (1:1 molar ratio) at room temperature for 30 min. The size and Zeta potential of InCasor probe were measured using a Zetasizer. Then, SEM and TEM imaging were utilized to study the morphology and element composition of InCasor.

To analyze the stability of InCasor probe, $InCasor_{LR}$ or $InCasor_{CLR}$ were separately incubated with Exonuclease I (Exo I, Thermo Fisher Scientific), 10% FBS, pH=7.4 PBS, pH=6.5 PBS, pH=5.0 PBS respectively at 37 °C overnight. Fluorescence quantification of reaction mixtures in 20 μL volume was performed using Synergy H1 Hybrid Reader (BioTek, USA) ($\lambda_{ex}$: 488 nm, $\lambda_{em}$: 520 nm).

**Preparation and characterization of SNA_DNF.** Thiolated segment strands were treated with 0.1 M DTT in disulfide cleavage buffer (170 mM PBS, pH = 8.0) for 2-3 hours at room temperature to reduce the disulfide protecting group to thiol. The oligo concentration was determined by UV-Vis. Then, ~3 μM (final concentration) thiolated oligonucleotides were mixed with ~8 nM AuNPs (final concentration) in DEPC-treated water and incubated for 2 h in the dark at -20°C. Next, any unbound ssDNA was eliminated through centrifugation (13,000 g, 4°C for 30 minutes). The precipitate was suspended in the washing buffer (comprising 10 mM PBS and 0.1 M NaCl, pH=7.4), and then subjected to another centrifugation. The resulting precipitate was resuspended in DEPC-treated $H_2O$ after collected by centrifugation. Then the $SNA_{DNF}$ were analyzed by ultraviolet absorption spectrum, agarose gel electrophoresis and TEM. The preparation and characterization of $SNA_{RNP}$ is similar to $SNA_{DNF}$. They both stored in the dark at -4°C.

**Preparation and characterization of InCasor-SNA_DNF.** The DNF and $SNA_{DNF}$ (1: 20 molar ratio) were mixed in PBS. Following a one-hour incubation at room temperature, any unbound $SNA_{DNF}$ was eliminated by subjecting the mixture to centrifugation at 12,000 g for 20 minutes. The hybridization of DNF and $SNA_{DNF}$ were analyzed by agarose gel electrophoresis. The DNF/$SNA_{DNF}$ were incubated with Cas12a/crRNA (1:1 molar ratio) for 30 minutes (RT) to form InCasor-$SNA_{DNF}$. Then, TEM imaging was employed to examine the morphology of InCasor-$SNA_{DNF}$.

### Cell lines and cell culture

The cell lines utilized in this study were procured from ATCC. MDA-MB-231 cells (HTB-26) were cultured in Leibovitz's L-15 (Procell, China) with 10% (v/v) fetal bovine serum (FBS, Lonsera, Suzhou Shuangru Biotechnology Co.,Ltd) and 1% penicillin/streptomycin (P/S) in an atmosphere at 37 °C without $CO_2$. HepG2 cells (HB-8065), MCF-7 cells (HTB-22), and A549 cells (CCL-185) were cultivated in DMEM(H) (Solarbio, China) with 10% FBS, along with 1% P/S. These cell lines were maintained in a 5% $CO_2$ atmosphere at 37 °C.

### Analysis of subcellular organelle localization of InCasor

To analysis the impact of Sgc8 aptamer for enhancing endocytosis of InCasor, HepG2 cells were exposed to 50 nM InCasor probe labelled with FAM with or without Sgc8 aptamer at 37 °C for 4 h. After staining by Hoechst 33342 (Beytime, China) and washing 3 times via PBS, the cells were subjected to fluorescence imaging performed on a Leica CLSM equipped with 63× oil objective. Subsequently, the cells were harvested for flow cytometry analysis. The acquired data were then analyzed using Flow Jo software. The gating strategy for cells can be found in Supplementary Fig. 20.

To explore the subcellular movement of InCasor probe, cells were exposed to 50 nM FAM- InCasor for 0 or 8 h. 50 nM LysoTracker Red (Beytime, China) was then introduced to cells for 30 minutes, and the cells were characterized using CLSM after Hoechst 33342 staining.

To assess the impact of Cyt C aptamer in InCasor on targeting of mitochondria, FAM- InCasor w/wo Cyt C apt (50 nM) were exposed to cells for a 6-hour duration. After which, the cells were incubated with Mito-Tracker Red (50 nM, Beytime, China) for 20 minutes and visualized using CLSM following Hoechst 33342 staining.

To examine the Cyt C apt-mediated entry of the Cas12a RNP in mitochondria, Cy5-NHS was employed to label the Cas12a protein, and FITC-NHS employed to mark crRNA. The Cy5-labeled Cas12a protein was produced by conjugating Cy5-NHS ester to the remaining primary amine groups on Cas12a protein as previously reported[31]. To prepare FITC-labeled crRNA, amino-modified crRNA was synthesized by in vitro transcription using amino-modified UTP and DNA template of full-length crRNA. After purification using TRIzon Reagent, it was incubated with FITC-NHS ester in borate buffer (pH = 8.5) for 3 h. FITC-labeled crRNA was obtained by ethanol precipitation method.

Cy5 labelled Cas12a and FITC-crRNA were employed to form the InCasor probe with or without Cyt C aptamer. HepG2 cells were then exposed to these probes for 6 h. After the incubation, Mito-Tracker Red and Hoechst 33342 staining were performed as mentioned above. The cells were then visualized using CLSM. The mitochondrial fraction was collected and subjected to flow cytometry analysis. The data were analyzed using Flow Jo. The gating strategy can be found in Supplementary Fig. 20.

For further investigation of the subcellular distribution of InCasor using Bio-TEM, 50 nM SNA$_{DNF}$/ SNA$_{RNP}$ labelled InCasor$_{Cyt\ C\ apt+}$ and InCasor$_{Cyt\ C\ apt-}$ probe were exposed to HepG2 cells for 6 h. After that, cells were fixed, sectioned (70 nm, Leica UC7, LKB 11800 PYRAMITOME) and examine in TEM (HITACHI, HT7700).

To quantitatively assess the penetration of InCasor into mitochondria, proteins, RNA, and DNA were extracted from mitochondria of HepG2 cells. HepG2 cells were exposed to either the 50 nM InCasor$_{Cyt\ C\ apt+}$ or InCasor$_{Cyt\ C\ apt-}$ for 6 h. After that, crude mitochondrial preparations were obtained as previous reported[22]. The isolated mitochondria were collected through centrifuge at 3500 g at 4°C for 10 minutes.

Protease protection assays were performed as previously reported. In brief, crude mitochondrial fractions (20 μg) were processed with proteinase K (50 μg/mL) on ice for 30 min to eliminate proteins located outside the mitochondria. The proteinase K reaction was halted by treating the mixture with 1 mM PMSF, and purified mitochondria were then collected. The mitochondrial proteins were extracted using RIPA lysis buffer and quantified using the BCA protein quantification kit.

For the RNase protection assay, crude mitochondrial fractions (20 μg) underwent treatment by 0.5 mg/mL RNase A (NEB) (37 °C, 30 min) to eliminate RNA situated outside the mitochondria. Purified mtRNA was subsequently extracted by TRIzon.

In the DNase protection assay, crude mitochondrial extract (20 μg) was mixed with 0.5 U/μL DNase I (37 °C, 15 min) to eliminate extra-mitochondrial DNA, and then purified mtDNA was extracted.

SDS PAGE gel was employed to examine the protein profiles extracted from mitochondria of HepG2 cells that underwent various treatments. Protein samples were denatured in 1× SDS-PAGE protein sample upload buffer (100°C, 5 minutes) and analyzed by Western blotting with 25 μg. After dilution into the same concentration, the samples were subjected to separation using an SDS-PAGE gradient gel. Following separation, they were stained by Cormas bright blue staining solution (15 min, RT). After 4-24 hours of decolorization at room temperature, image the protein bands on a Bio Spectrum® 615 Imaging System.

Previously reported RCA-based and Cas13a-based RNA assays were used to quantify crRNA in InCasor through Cyt C aptamer-mediated entry into mitochondria[22]. 0.8 μg mtRNA were firstly incubated with 5 'phosphorylated Template-crRNA in 1× T4 DNA Ligase buffer with T4 DNA ligase (Thermo Fisher Scientific). The RCA reaction was conducted as previously reported. The RCA product was analyzed by 1% agarose gel electrophoresis.

CRISPR/Cas13a system was employed to sensitive detect 5'crRNA$_{ND4}$ from mtRNA of HepG2 cells. Specifically, 100 nM Lwa-Cas13a, 150 nM crRNA of Cas13a, 0.8 μg total RNA, 0.5 μM Poly U Reporter were mixed in 1× Reaction buffer. Following a 30-minute incubation at 37 °C, the fluorescence signal was analyzed ($\lambda_{ex}$: 488 nm; $\lambda_{em}$: 520 nm).

To assess quantitatively the entrance of InCasor into mitochondria, PCR was used to analyze specific DNA fragment of DNF in InCasor probe. The thermal cycling conditions were as follows: 94°C for 3 minutes; 45 cycles of 94°C 30 s, 60°C 30 s, and final extension 72°C for 5 minutes. The resulting products were characterized using a 2% agarose gel.

To assess intracellular $Mg^{2+}$ release induced by InCasor, the HepG2 cells treated with 50 nM InCasor for 6 h were digested using trypsin-EDTA solution, and collected by centrifugation (200 g for 5 min). The concentration of $Mg^{2+}$ in HepG2 cells was measured by ICP-MS (Agalient, 7800) after cell counting.

To analyze the change of intracellular $Mg^{2+}$ levels in a long time, HepG2 cells were treated with 50 nM InCasor for different times (0, 2, 4, 6, 8, 10, 12, 24, and 36 h), respectively. After 8 h of incubation, the InCasor-containing medium was replaced with fresh medium for longer time culture. After treated with 50 μM Mg-Fluo-4 AM for 20 min and Hoechst 33342 staining, the cells were imaged using CLSM. The cells were then gathered and analyzed by ICP-MS using an Agilent 7800.

## Fluorophore-quencher ssDNA cleavage assays

50 nM Cas12a with 75 nM crRNA in 1× reaction buffer were incubated at 37 °C for 10 minutes to form Cas12a/crRNA RNP complexes. The reaction was initiated by adding 50 nM activator dsDNA and 1 μM short ssDNA reporter (FAM-TTATT-BHQ-1) in a 20 μL reaction. Analysis of divalent cation preference of Cas12a were performed as above, with the following modifications: the 10 mM $MgCl_2$ in 1× reaction buffer was replaced by equal concentration of $ZnCl_2$, $CuCl_2$, $CoCl_2$, $FeCl_2$, $CaCl_2$. Following a 30-minute incubation at 37 °C, the fluorescence signal was determined ($\lambda_{ex}$: 488 nm; $\lambda_{em}$: 520 nm).

## Cell viability analysis

**CCK8 assay.** Cell Counting Kit-8 (CCK8) assay was employed to evaluate cell viability after treatment with the InCasor. Specifically, $1\times10^4$ HepG2 cells were seeded on 96-well plate per well overnight. After the cells were incubated with 50 nM InCasor probe for 6 h and washed with PBS. Detection was carried out at absorption wavelength of 450 nm.

**Cell cloning assay to detect cell viability.** Take the monolayer cultured cells treated with the InCasor probe, they were digested with 0.25% trypsin and resuspended as single cell suspensions using culture medium with 10% FBS. Dilute the cell suspension in multiples of gradient, inoculate it in a dish with 10 mL pre-warmed culture medium, and rotate it gently to disperse the cells evenly. Place it in an incubator at 37 °C with 5% $CO_2$ for 2 to 3 weeks. Culture was stopped when macroscopically visible clones appeared in petri dish. Use 0.1% crystal violet staining solution to stain and visualize the number of clones greater than 10 cells under the microscope. Finally, 33% glacial acetic acid was used to dissolve the crystal violet at 37 °C, and the cell viability was calculated by measuring absorbance at 590 nm.

## InCasor for in vitro detecting mtDNA

In vitro mtDNA detection protocol utilizing InCasor is as follows, unless otherwise specified: 250 nM InCasor, 250 nM dsDNA were mixed in 1× Tolo buffer. Following a 30-minute incubation at 37 °C, the fluorescence signal was quantified ($\lambda_{ex}$: 488 nm; $\lambda_{em}$: 520 nm).

To analysis of self-cleavage kinetics of InCasor probe induced by target dsDNA, the same reaction mixture mentioned above was incubated at 37 °C for various time points, and the signal of each reaction mixture was analyzed previously described.

For isolated mitochondria genotype analysis in vitro using InCasor, 250 nM InCasor probe, 5 µg mitochondria (MT or WT) were mixed in 1× Tolo buffer. Following a 2 h incubation at 37 °C, the signal was analyzed previously described.

## Analysis of self-cleavage of InCasor

250 nM InCasor, 250 nM dsDNA were added to 1× TOLOBIO buffer 3. After incubated at 37 °C for 30 minutes, the reaction mixtures were analyzed by 1% agarose gel electrophoresis. TEM imaging was employed to investigate the morphology of InCasor before and after incubation with target dsDNA.

To analysis of self-cleavage kinetics of InCasor probe induced by target dsDNA, the same reaction mixture mentioned above was incubated at 37 °C for various time points interval and analyzed by 1% agarose gel electrophoresis.

## Imaging of mtDNA in live cells using InCasor probe

Cells were cultured overnight the day and incubated by 50 nM InCasor for 6 h at 37 °C. Hoechst 33342 and Mito-Tracker Red were used to stain cell nucleus and mitochondria respectively before imaging. Cells were fluorescence imaged on a Leica CLSM equipped with 63× oil objective. The above cell samples were then analyzed using flow cytometry.

For analysis of the stability of InCasor probe in cell culture environment, InCasor$_{NT-LR}$ or InCasor $_{NT-CLR}$ were separately incubated with 10% FBS respectively at 37 °C overnight and then incubated with HepG2 cells, followed by confocal imaging.

For analysis of mtDNA genotype in mixed isolated mitochondria in vitro using InCasor probe, the isolated MT-Mito were pre-stained with CellMask™ (Green, Invitrogen™) after extraction from MDA-MB-231 cells. Similarly, the mitochondria with wild-type mtDNA (WT-Mito) from recipient cells (HepG2) were labeled by Mito-Tracker (Red). After quantification of mitochondria by the BCA protein quantification kit, WT-Mito and MT-Mito were mixed in mass ratio 1:1. Then 250 nM InCasor probe, 5 µg mixed mitochondria were added to 1× TOLOBIO buffer 3. After incubation at 37 °C for 2 h, 10 µL reaction mixture was is dropped on adhesive slide. After standing for 10 minutes, the fluorescence signal was observed with confocal microscope.

For analysis of mtDNA genotype in co-cultured live cells using InCasor probe, the HepG2 cells were pre-stained with DiI ($\lambda_{ex}$: 552 nm; $\lambda_{em}$: 580 nm, Beytime, China) and the MDA-MB-231 cells were pre-stained with DiD ($\lambda_{ex}$: 638 nm; $\lambda_{em}$: 680 nm, Beytime, China) before cocultured at 37 °C overnight. The cells were incubated by 50 nM InCasor ($\lambda_{ex}$: 488 nm; $\lambda_{em}$: 520 nm) for 6 h, followed by confocal imaging.

## Western blotting

Western blotting was employed to characterize the level of ND4 expression in different samples. To evaluate the gene editing efficiency of InCasor, HepG2 cells were exposed to 50 nM DNF/CLR, InCasor$_{NT}$ or InCasor$_{ND4}$ overnight. Proteins were then collected using lysis buffer and quantified. Protein samples denatured in 1× SDS-PAGE protein sample loading buffer (100°C, 5 minutes) were analyzed by Western blotting with 25 µg. After dilution into the same concentration, these proteins were isolated using SDS-PAGE gradient gel and transferred to PVDF membrane, and then blocked with QuickBlock™ Block Buffer (Beyotime, China). The ND4 (1:50, Abcam, Cat# ab116897) and GAPDH (1:1000, Abcam, cat: ab128915) primary antibodies along with HRP-coupled anti-mouse IgG secondary antibody (1:10,000 in TBST; Proteintech, Cat#SA00001-1) and an anti-rabbit IgG secondary antibody (1:10,000 in TBST; Proteintech, Cat#SA00001-2). The membrane was finally detected on Bio Spectrum® 615 imaging System.

For analyzing the entrance of Cas12a in InCasor to mitochondria, RIPA lysate was used to collect mitochondrial proteins after extracting mitochondria from HepG2 cells with different treatment, and BCA protein quantification kit was used for quantification. The procedures for Western blot assay were the same as above. The antibodies used in this experiment is LbCpf1/Cas12a Mouse mAb (Strain ND2006, 2D5-6G11, Cell Signaling Technology), MFN2 Polyclonal antibody (Proteintech, Cat #12186), HSP 60 (A57-B9, Santa Cruz Biotechnology, Cat# sc-57840) and Rabbit monoclonal anti-GAPDH antibody (14C10, Cell Signaling Technology).

## qPCR assay

Due to the low efficiency of homologous and non-homologous repair for mtDNA[56], the broken mtDNA tends to be rapidly degraded[57]. Therefore, T7E1 assay, Sanger sequencing, and NGS are not suitable for evaluating the efficiency of mitochondrial gene editing[58,59]. Thus, qPCR was used in this work to analyze mtDNA gene editing by InCasor in different samples. To analyze the cutting of mtDNA triggered by InCasor, 50 nM of DNF/CLR, InCasor$_{NT}$ and InCasor$_{ND4}$ were added to cultured HepG2 cells. After incubation for 6 h, the cells were rinsed with PBS. The DNA was extracted by TIANamp Genomic DNA Kit. Quantitative PCR was conducted on Light Cycler®96 System. The sequences of primers are shown in Supplementary Table 1.

## InCasor for imaging the heteroplasmic mtDNA mutations in live cells

**Mitochondria isolation.** To prepare cell line with heterogenous mtDNA, we isolated mitochondria with mutant mtDNA (MT-Mito) from MDA-MB-231 cells and transfer to HepG2 cells. After digestion of MDA-MB-231 cells with trypsin-EDTA solution, the cells were collected by centrifugation (200 g for 5 min). Following cell counting, adding 1 mL of mitochondrial separation reagent to a batch of 20 million cells, gently suspend and place in ice bath for 10 min. After fully grinding in a low-temperature tissue homogenizer (Servicebio, China), the cells were centrifuged at 1000 g at 4°C for 10 minutes. Carefully transfer the supernatant to another centrifuge tube and the isolated mitochondria were collected through centrifuge at 3500 g at 4°C for 10 minutes. RIPA lysate was used to collect mitochondrial proteins, and BCA protein quantification kit was used for mitochondria quantification.

**Transfer to recipient cells and live cell imaging.** The isolated MT-Mito were pre-stained with CellMask™ (Green, Invitrogen™) after extraction from MDA-MB-231 cells. Likewise, the mitochondria with wild-type mtDNA (WT-Mito) in recipient cells (HepG2) were labeled by Mito-Tracker (Red). Before transferring MT-Mito to HepG2 cells, the amount of MT-Mito (µg of protein) was adjusted according to the results of BCA protein quantification (for Fig. 5c-d, the amount of mitochondria were 1.0 µg; for Fig. 5f, the amount of mitochondria were 0, 0.1, 0.25, 0.5, 0.75, 1.0 µg). Discard the original medium from the HepG2 cells, and rinse the cells with PBS. The mitochondrial suspension was slowly added near the bottom of the plate, followed by adding 200 µL DMEM/FBS (5%). After incubated overnight at 37°C, the cells were rinsed by PBS and treated by 50 nM InCasor probe at 37°C for 6 h, followed by confocal imaging.

**Mitochondrial membrane potential (Δφm) measurements.** Δφm of isolated MT-Mito was assessed by JC-1 assay kit (Beyotime Biotechnology). 20 µg isolated MT-Mito were incubated with JC-1 probe for 10 min at 37 °C. The signal was then analyzed ($\lambda_{ex}$: 488 nm, $\lambda_{em}$: 520 nm; $\lambda_{ex}$: 552 nm, $\lambda_{em}$: 590 nm).

## Tumor model in animal experiment

5-weeks-old BALB/c nude mice (female) were acquired from Hunan SJA Laboratory Animal Co., Ltd. These mice were then housed under controlled conditions, with a temperature of 25°C and humidity maintained at 55% in Experimental Animal Center of Zhengzhou University. Animal experiments were conducted according to the guidelines and regulations approved by the Henan Laboratory Animal Center. HepG2 cell tumor-bearing mice were achieved by subcutaneous injection of $1 \times 10^7$ HepG2 cells into female BALB/c nude mice. MDA-MB-231 cell tumor-bearing mice were achieved by subcutaneous injection of $1 \times 10^7$ MDA-MB-231 cells into female BALB/c nude mice. To construct a mouse model of dual tumor with HepG2 cells and MDA-MB-231 cells, we injected HepG2 cells sub-cutaneously into the left forelimb of female BALB/c nude mice, and MDA-MB-231 cells were injected subcutaneously into the right forelimb. During the course of the in vivo experiments, the mice were closely monitored on a daily basis and euthanized if any of the mice exhibited a weight loss exceeding 20%. The maximum diameter of the tumor volume in the mice did not surpass 15 mm, with the approval of the Henan Provincial Animal Experimentation Ethics Committee.

## Biodistribution analysis

To calculated the half-life of InCasor, mice were injected i.v. with Cy5-ssDNA and InCasor (equivalent concentration: 10 nmoles, 200 μL). Blood samples were collected at various time points: 0.5, 1, 2, 4, 6, 8, 10, 12, and 24 hours after injection (n = 3). The samples were diluted with 80 μL PBS and the fluorescence intensity of Cy5 was determined ($\lambda_{ex}$: 638 nm; $\lambda_{em}$: 680 nm).

After a 14-day implantation period, mice were injected with Cy5-labelled ssDNA, InCasor$_{Sgc8-}$, or InCasor$_{Sgc8+}$ (10 nmoles, 200 μL). Subsequently, whole-body images were captured at 2, 4, 6, 12, and 24 h after the injections, which were acquired using Bruker In-Vivo Xtreme ($\lambda_{ex}$: 640 nm; $\lambda_{em}$: 710 nm). Ex vivo imaging of the Cy5 signal within tumor, heart, liver, spleen, lung, and kidney were conducted using the Bruker In-Vivo Xtreme imaging system after mice were euthanized with $CO_2$. The section slides were imaged by CLSM.

## Imaging of mtDNA mutation in vivo

After a 14-day implantation period, mice were intertumoral injected with 0.5 nmole, 50 μL or tail-vein injected with 1 nmole, 200 μL InCasor with different crRNA respectively. Subsequently, whole-body images were captured at 1, 2, 4, and 6 hours after the injections, which were acquired using Bruker In-Vivo Xtreme ($\lambda_{ex}$: 640 nm; $\lambda_{em}$: 710 nm). Ex vivo imaging of the Cy5 signal within umor, heart, liver, spleen, lung, and kidney were conducted using the Bruker In-Vivo Xtreme imaging system after mice were euthanized with $CO_2$. The section slides were imaged by CLSM. Unless otherwise stated, the standard procedure for InCasor probes for in vivo imaging of animals was as described above.

## Safety evaluation

To assess the biological safety of InCasor, a series of evaluations and tests were conducted: blood samples were collected from the eyeballs for hematology and blood biochemistry tests; major organs of mice were imaged by HE staining to assess systemic safety; tumor tissues were stained using a TUNEL detection kit (Roche, Shanghai, China) to assess their tumor damage capacity.

## Immunofluorescence Staining Assay

The tumor tissue sections were treated with 3% bovine serum albumin solution (1 h, RT), and then de-graphitized and antigen retrieval. Arginase-1 (D4E3M™) XP® Rabbit mAb (CST, USA) were incubated at 4°C overnight.

## Sequencing of mtDNA

In this study, genomic DNA from the specified cell lines was extracted as mentioned above. PCR reactions carried out on a BioRad T100 thermal cycler. The products were purified by gel extraction and analyzed using Sanger sequencing.

## Pyrosequencing

The mt.13105 A > G mtDNA heterologous genes were evaluated by pyrosequencing. PCRs were produced with template DNA (50 ng) using the SanTaq PCR kit as follows: 95 °C for 5 minutes; 35 cycles of 94°C 30 s, 55°C 30 s, 72°C 50 s, and 72°C for 8 minutes. The primers were shown in Supplementary Table 1. The NCBI reference sequences NC_012920.1 were employed for the human tumor cell line mitochondrial genomes.

## Whole-genome sequencing

A Rapid Animal Genomic DNA Isolation Kit was employed to extracted genomic DNA of HepG2 cells. The purity, quantity and size of genomic DNA were evaluated using agarose gel electrophoresis and NanoDrop. To perform high-throughput sequencing on the Illumina platform, whole-genome DNA library preparation was carried out in GENEWIZ with a mean coverage of 40.5-, 50.6- and 46.2-fold for NC, InCasor$_{NT}$ and InCasor$_{ND4}$ group, respectively. The Q30 was set to be above 90%, and the required average error rate to be was below 0.1%. The generated sequencing data were then aligned to the hg19 reference genome using the Genome Analysis Toolkit 4.1.1.0. The extraction process involved identifying all polymorphic Single Nucleotide Variant (SNV) and insertion/deletion (indel) sites in the genome. Subsequently, high-confidence SNV and indel datasets were generated and subjected to in-depth analysis.

## Microscopy

Microscopic imaging conducted on a LEICA TCS SP8 STED inverted microscope equipped with various objectives, including a 100× PLAN APO oil objective with a numerical aperture (NA) of 1.9, a 60× PLAN APO oil objective with an NA of 1.4, and a 40× PLAN APO oil objective with an NA of 1.3. The imaging system also featured an Evolve Delta 512 EMCCD camera and lasers with wavelengths of 405 nm, 488 nm, 552 nm, and 638 nm. Image acquisition was managed through LAS AF Lite software. There are three main types of fluorescent dyes used in cell imaging: (1) for nucleic acid labeling by chemical modification, (2) commercially available fluorescent probes for organelle labeling, (3) for protein labeling by covalent attachment. The specific λex and λem of the fluorescent dyes utilized in this work can be found in Supplementary Table 2.

## Image processing and analysis

Image processing was carried out using Fiji, an open-source image analysis software. A single microscope plane displaying the maximum fluorescence of mitochondria or a projection generated from adjacent Z planes that displays the maximum fluorescence at the selected loci is presented. To enhance image quality for optimal visualization, adjustments were made using the brightness/contrast function in Fiji, which involved linearly modifying the maximum and minimum values. Relative fluorescence measurements were obtained from setting the mean fluorescence intensity of background in NC. PCC determines the pixel-by-pixel covariance of two images and ranges from -1 to 1. -1 indicates negative correlation and 1 indicates full colocalization. The analysis of PCCs was conducted using coloc2 plugin.

## Statistics and reproducibility

Unless otherwise specified, all the experiments were conducted independently three times at least. The number of independent replicate

samples (n) and statistical parameters are provided in the legends corresponding to the specific experiment. Sample sizes were not determined using statistical methods before the experiments. Graph-Pad Prism software was used to perform statistical analyses. Two-tailed unpaired Student's t-tests were utilized for two-group comparisons., one-way analysis of variance (ANOVA) followed by post hoc Tukey's tests was performed for multiple-group comparisons ($*P < 0.05$, $**P < 0.01$, $***P < 0.001$ and $****P < 0.0001$; ns: no significance).

### Reporting summary
Further information on research design is available in the Nature Portfolio Reporting Summary linked to this article.

## Data availability
The next-generation sequencing data produced in this study have been submitted and can be accessed in the BioProject database under accession code SRR18056313, SRR18058193, and SRR18056642. The authors declare that all data generated or analyzed during this study are included in this published article and available from the Figshare (https://doi.org/10.6084/m9.figshare.24458674). Source data are provided with this paper.

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

## Acknowledgements

This work was supported by funds to K.Z. from the National Natural Science Foundation of China (No. 22122409, U2004197); Henan Province Fund for Cultivating Advantageous Disciplines (No. 222301420019); Programs for Science & Technology Innovation Talents in Universities of Henan Province (No. 21HASTIT043). All the animal experiments were performed in accord with the guidelines of the Regional Ethics Committee for Animal Experiments and Zhengzhou University Institutional Animal Care and Use Committee. The authors thank Modern Analysis and Computing Center of Zhengzhou University for technical assistance.

## Author contributions

K.Z., J.S., and R.D. conceived and designed the experiments. Y.L., R.X., Y.W. J.G., D.H., Y.P., F.Q., and Y.Z. performed the experiments. Y.L., K.Z., R.D., J.L, W.L, and H.G. analyzed the results. Y.L., and K.Z. wrote the manuscript. K.Z., J.S., R.D., and Z.Z. supervised the entire project.

## Competing interests

The authors declare no competing interests
