## [Peer Review File · Nature Communications]

Reviewers' comments:

Reviewer #1 (Remarks to the Author)

The authors developed a CRISPR-based sensor (InCasor) to detect mtDNA mutations by in vivo imaging. They found that the delivery of InCasor which consists of Cas12a/crRNA complex, fluorescence reporters and DNA-Mg²⁺ hybridized nanoflower enables recognition and imaging of mtDNA mutations. InCasor has been tested to work in different cell lines and mouse models to allow mtDNA SNV imaging.

Major:

1. The significance of InCasor is limited since the SNV has to be within or adjacent to the PAM sequence to be detected.
2. InCasor potentially cuts to destroy the target DNA sequence while detecting the SNVs. Is there any way to avoid this side effect?

Minor:

Unclear description of Fig. 1e. What do STEM and EDS stand for in the figure legend?

Reviewer #2 (Remarks to the Author)

The authors have shown very interesting approach to map out mitochondria DNA mutations using cas12 based approach. The work is of excellent quality, however, I have some basic queries which I believe must be addressed before publication.

1. Please talk about mtDNA heterogeneity and how it complicates the acquired mutation detection in the introduction part.
- 2.2. How stable InCasor is when (i) it encounters the target DNA and (ii) when there is no target DNA, in-vitro?
3. Characterization of InCasor in-vivo is missing. How stable it is in-vivo and how it is expelled from the circulatory system?

4. As per my understanding PTK7 is NOT expressed on all adult cells, kindly explain the rationale and provide suitable references.
5. Since Mg²⁺ is involved in regulation of ion channels, binds to ATP and affects mitochondria, how does authors ensure that excessive Mg²⁺ has no long term effects on the cells. The authors state that
6. Since Mg²⁺ is already in high concentrations inside the cells and mitochondria, did authors tested InCasor activity with providing Mg²⁺ externally. Please provide data in relation to cellular Mg²⁺.
7. What is the sensitivity of the mutation detection system, Can it identify 1 mutations in one of the mitochondria? What is the estimated minimum number of mutations required? Can the signal be quantified?
8. Sensitivity of in-vivo imaging using fluorochromes has very limited usability.
9. Please show self degradation/half life on InCasor inside the mitochondria.
10. Please share data regarding non-specific localization of InCasor in cytoplasm and/or nucleus.

Reviewer #3 (Remarks to the Author)

Overview:

The authors propose a novel application of Cas12a trans-cleavage combined with DNA nanoflower for single nucleotide variation detection in live cells. They call this system, "InCasor". Through a series of controlled assays in hepatocytes, the authors detected efficient cleavage of a circular DNA reporter and uptake of DNFs into mitochondria. They further demonstrate concentration dependency of detection with a threshold of <10nM InCasor, high mtDNA target specificity with essentially no nuclear off-target activity, and compatibility across multiple cell lines. Additionally, the authors were able to engineer the InCasor system to detect SNVs outside canonical PAM sites and showed highly specific detection of mutant mitochondria in a laboratory-generated heteroplasmic cell line. Finally, in a mouse model, injected InCasor was able to specifically localize to a tumor site only in the presence of the Sgc8 aptamer (targeting PTK7 receptor) and engineered crRNAs targeting mutations with one or two mismatches.

General comments:

Overall, this is a well written account of a detailed study, utilizing a wide array of techniques, describing the potential use of Cas12a (InCasor) for the detection of single mutations in mitochondrial DNA that are

often associated with human disease states, including metastatic cancers. The individual experiments are well-conceived, controlled, and clearly presented. Collectively, they suggest that the InCasor has potential utility for SNV detection across a wide range of diagnostic applications. In my opinion, the study is highly relevant as a proof-of-concept and would be interesting to the general audience of the journal. I offer the following suggestions for revision for the authors to consider.

1. The use of Cas12a as a detection system has expanded greatly over the past few years. The introduction would have more impact if it were expanded to provide additional context regarding the novelty of the current study, considering the breadth of Cas12a applications.

2. Most of the experiments and statistical analyses comprise only three replicates. Although this may be acceptable, the replicates also show remarkably similar magnitudes; i.e. little variance, which is somewhat surprising for what are presumed to be biological replicates. The authors should address the nature of the replicates in the Methods and mention this as a potential limitation to their outcomes in the Discussion.

3. The study is also restricted to mitochondrial genes N4 & N5 and a couple of SNVs. This should be included in the Discussion as another possible limitation to broad applicability; e.g. more genes and SNVs should be tested. Would the authors suggest particular genes/SNVs for follow on studies?

4. Because of the number of panels per figure, some of the images are quite small at normal magnification. For example, individual mitochondria in Figure 4f are difficult to see, especially as the color merge between MT-Mito and Cy3-InCasor looks very similar to the separate panels. Zooming helps, but an inset with a single mitochondrion at higher magnification would be warranted.

5. Sgc8- InCasor was highly localized to kidney and ND4-2-MT & ND5-MT-1 injected into the tail vein produced fluorescence outside the tumor region. The latter results seem to be explained as metastatic cells in the liver, but could these results indicate potential false-positives in other cells that weren't examined? Further, could false-positives arise if either the stability of the aptamer is compromised or via unrecognized gene-specific off-target effects with engineered crRNAs? Moreover, the Sgc8+ InCasor was not exclusively localized to the tumor. How could this be improved? These issues should be addressed in the Discussion.

6. Line 270: Remove "Single nucleotide variation"; defined in line 24

Responses to the comments of the reviewers

Response to reviewer 1:

The authors developed a CRISPR-based sensor (InCasor) to detect mtDNA mutations by in vivo imaging. They found that the delivery of InCasor which consists of Cas12a/crRNA complex, fluorescence reporters and DNA-Mg²⁺ hybridized nanoflower enables recognition and imaging of mtDNA mutations. InCasor has been tested to work in different cell lines and mouse models to allow mtDNA SNV imaging.

Response:

Thanks for your instructive suggestions. Based on your suggestions, we have added some new data. And all the modified portions were highlighted in gray.

Major

Comment 1:

The significance of InCasor is limited since the SNV has to be within or adjacent to the PAM sequence to be detected.

Response 1:

Thanks for your comment. As you mentioned, the widespread applicability of InCasor across various SNV sites is crucial to its significance, which is limited by the inherent mismatch tolerance of the CRISPR/Cas12a system, especially for the SNV sites away from PAM sequences. To solve this issue, several approaches have been investigated for improving the SNV recognition capability of the CRISPR system, such as truncating the spacer region of the crRNA (*Nat Commun.* 2021; 12(1):1739.), chemically modifying the crRNA (*Chem. Sci.*, 2022, 13,2050.), introducing hairpin structures (*Adv. Sci.* 2021, 2003611; *Nat Biotechnol.* 2019; 37(6):657-666.) etc., to achieve higher specificity for the recognition of disease-associated key point mutations. In this work, we used insertion of mismatches to modify crRNAs thereby achieving higher recognition specificity for SNV adjacent to the PAM sequence.

According to the reviewer's comment, we tried to further engineer the crRNA and tested the ability for recognition of SNV far away from PAM sequences. Specifically, we selected two loci with mutations at position 12 or 16 nt away from PAM,

respectively, and modified the crRNA to improve the recognition specificity. mt.4769A>G, located in *ND2* gene, was selected based on the sequencing results of MDA-MB-231 used in this study. mt.3916G>A, located in *ND1* gene, was selected according to the literature (*Hum. Mutat.* 2020, 41, 2028–2057; *Nat Rev Cancer.* 2021, 21(7):431-445.), which was related to oncocyoma. For each mutation site, we designed five crRNAs separately: fully matched with mutant site, named crRNA_{FM}; inset single mismatch near PAM, named crRNA_{SM1}; inset single mismatch near SNP site, named crRNA_{SM2}; inset double mismatches near PAM, named crRNA_{DM1}; inset double mismatches near SNP site, named crRNA_{DM2}. After synthesizing crRNAs through *in vitro* transcription, an *in vitro* fluorescence analysis method (as described in **Methods**) was employed to distinguish between wild-type and mutant mtDNA. And the discrimination factors (DFs) were calculated. The results are shown below:

Figure R1. Engineering crRNA for recognition of mutations in mtDNA that are distant from the PAM. a. Sequences of crRNA_{FM} (fully matched with the 4769A>G mutant *ND2* gene) and engineered crRNAs (with different numbers of nucleotide mismatches at different position). Each SNV is indicated by a bold red letter. Each mismatched position is indicated by an underlined bold blue letter. **b.** An *in vitro* fluorescence assay was used to estimate the ability of different InCasor probes to detect the 4769A>G mutation in the *ND2* gene. n=3, data show mean ± SD. **c.** The discrimination factors of the InCasor probe with different crRNAs toward WT and MT mtDNA at 4769A>G. **d.** Sequences of crRNA_{FM} (fully matched with the 3916G>A mutant *ND1* gene) and engineered crRNAs (with different numbers of nucleotide mismatches at different position). Each SNV is indicated by a bold red letter. Each

mismatched position is indicated by an underlined bold blue letter. **e.** An *in vitro* fluorescence assay was used to estimate the ability of different InCasor probes to detect the 3916G>A mutation in the *NDI* gene. n=3, data show mean \pm SD. **f.** The discrimination factors of the InCasor probe with different crRNAs toward WT and MT mtDNA at 3916G>A. n=3, data show mean \pm SD. The sequences of all crRNAs used are listed in Supplementary Table 1.

As shown in **Figure R1**, when the mutation site is distant from the PAM sequence, the fully matched crRNA and crRNA_{SM1} cannot distinguish between wild-type and mutant mtDNA. In contrast, the insertion of two discrete mismatches in the seed region (crRNA_{DM1}) and the insertion of 1 mismatch in the SNP locus (crRNA_{SM2}) both showed improved DFs (for mt.4769A>G, DF of crRNA_{SM2} was 4.1, DF of crRNA_{DM1} was 3.6; for mt.3916G>A, DF of crRNA_{SM2} was 2.4, DF of crRNA_{DM1} was 2.2). According to the results, engineering the crRNA could improve the ability of InCasor for sensing SNP sites distant from the PAM sequence.

In addition, researchers have designed other innovative strategies to achieve specific detection of SNV without the need of PAM sequence. These strategies involve the introduction of auxiliary probes, such as the toehold-activator (*Nucleic Acids Res.* 2022, 50(20):11727-11737.) and the use of sticky-end dsDNA (*Nucleic Acids Res.* 2022, 50(22):12674-12688.), which provide new ways for sensing almost all kinds of SNV sites.

According to your suggestion, the following text has been added to the **Results** section in the revised manuscript (Page 19) to clarify the ability of InCasor to recognize single-base mutation sites far from the PAM sequence.: “To further examine InCasor's ability to recognize SNVs distant from the PAM sequence, we selected two loci with mutations at position 12 or 16 nt after PAM according to sequencing results or literature^{40,41}, respectively, and modified the crRNA to test its recognition by InCasor. Compared to fully matched crRNA and crRNA_{SM1}, the insertion of two discrete mismatches in the seed region (crRNA_{DM1}) and the insertion of 1 mismatch in the SNP locus (crRNA_{SM2}) both showed similarly improved discrimination factors (DFs) (for mt.4769A>G, DF of crRNA_{SM2} was 4.1, DF of crRNA_{DM1} was 3.6; for mt.3916G>A, DF of crRNA_{SM2} was 2.4, DF of crRNA_{DM1} was 2.2) (**Supplementary Fig. 12**). Overall, engineering the crRNA contributed to InCasor's ability to sense SNV sites distant from

the PAM site.”. And the following text has been added to the **Discussion** section in the revised manuscript (Page 30): “Moreover, the crRNA engineering process substantially enhanced Cas12a’s ability for recognition of single-base mutations in non-PAM sequences.”

And Figure R1 has been amended as **Supplementary Fig. 12.** in the revised **Supplementary Information** (Page 27).

40 Kopinski, P. K., *et al.* Mitochondrial DNA variation and cancer. *Nat Rev Cancer* **21**, 431-445, (2021).

41 McCormick, E. M. *et al.* Specifications of the ACMG/AMP standards and guidelines for mitochondrial DNA variant interpretation. *Human Mutation* **41**, 2028-2057, (2020).

have been added to **References** (Page 33) in the revised manuscript as marked with a gray background.

Comment 2:

InCasor potentially cuts to destroy the target DNA sequence while detecting the SNVs. Is there any way to avoid this side effect?

Response 2:

Thanks for your comment. The intrinsic cis-cleavage activity of Cas12a, serving as the central function of InCasor, inevitably leads to the cleavage of the target DNA sequence during SNV detection (*Science. 2018; 360(6387):436-439.*). According to the literature, several potential strategies may possibly be used to avoid this undesired side effect. For example, protein modification has been regarded as a way to develop new functions of Cas proteins (*Nucleic Acids Res. 2022; 50(22):12689-12701.; Adv. Sci. 2023, 2206517.*). As an instance, Cas9 maintains its specific DNA binding properties in the absence of nuclease activity, resulting in the development of the catalytically inactive variant, known as dCas9, which finds widespread utility in non-editing applications (*Nat Biomed Eng. 2023 Aug 7. doi: 10.1038/s41551-023-01078-2.*). Moreover, some other SpCas9 variants, namely, nCas9 (D10A) and nCas9 (H840A), which cleave target (guide RNA-pairing) and non-target DNA strands, respectively, are

widely used for various purposes, including paired nicking, homology-directed repair, base editing, and prime editing (*Nat Biotechnol.* 2018 Oct 1. doi: 10.1038/nbt.4261). Besides, the capacity of Cas14a1 has been reported to provide trans-cleavage of ssDNA, specifically activated by a target RNA, without cleaving the RNA sequence (*Angew Chem Int Ed Engl.* 2021; 60(45):24241-24247).

In addition, a diverse array of CRISPR-Cas systems continues to emerge, showing various functionalities (*Science* 2023; 380, 410–415; *Nucleic Acids Res.* 2023; gkad495.; *Nat Biotechnol.* 2023; 41(4):500-512.; *Science.* 2019; 363(6422): 88-91. *Nat Commun.* 2021; 12(1):555. *Nat Rev Microbiol.* 2020; 18(2):67-83.). Therefore, it may be possible to achieve the trans-cleavage activity of Cas protein without cleaving the target DNA by mutational modification of Cas12a or developing other CRISPR systems, thus enabling non-destructive sensing SNVs of mtDNA in living cells.

According to your suggestion, we revised as followed:

“Besides, InCasor potentially cuts to destroy the target DNA sequence while detecting, which inevitably affect the cell viability during the sensing process. The potential solution may be finding some new Cas effectors or engineering a new Cas12a protein, which can recognize target mtDNA and induce trans-cleavage, but won't cut target sequence.” was added to the **Discussion** section (Page 29) as marked with gray background.

Minor:

Comment 3:

Unclear description of Fig. 1e. What do STEM and EDS stand for in the figure legend?

Response 3:

Thank you for your comment. STEM stands for Scanning Transmission Electron Microscopy, where a focused electron beam is scanned over a sample. STEM is an advanced imaging technique used in the field of materials science and nanotechnology to obtain high-resolution images and detailed information about the structure and composition of materials. There are different modes and techniques within STEM

microscopy, such as high-angle annular dark field (HAADF) imaging, which provides compositional information based on the intensity of scattered electrons, and energy-dispersive X-ray spectroscopy (EDS), which can be used to analyze the elemental composition of the sample.

We have revised the figure legend of Fig. 1e, the following text has been amended to the **Results** section (Page 8) in the revised manuscript: “The scanning transmission electron microscopy (STEM) and energy dispersive spectroscopy (EDS) mapping of InCasor probe. Scale bar: 100 nm.”

Response to reviewer 2:

The authors have shown very interesting approach to map out mitochondria DNA mutations using cas12 based approach. The work is of excellent quality, however, I have some basic queries which I believe must be addressed before publication.

Response:

Thanks for your appreciation for our work with instructive suggestions. Based on your suggestions, we have added some new data. And all the modified portions were highlighted in turquoise.

Comment 1:

Please talk about mtDNA heterogeneity and how it complicates the acquired mutation detection in the introduction part.

Response 1:

Thanks for your suggestion. The polyploid nature of the mitochondrial genome, up to several thousand copies per cell, gives rise to a crucial aspect of mitochondrial genetics known as homoplasmy and heteroplasmy (*Genome Res.* 2009; 19(4): 576-580.). Homoplasmy is when all copies of the mitochondrial genome are identical; and heteroplasmy is when there is a mixture of two or more mitochondrial genotypes. mtDNA mutations are mainly caused by the high level of reactive oxygen species (ROS) generated during oxidative phosphorylation, and the lower-fidelity of DNA replication in mitochondria. Heteroplasmy of mtDNA can primarily occur through somatic

mutagenesis during an individual's lifetime, which are mainly single-nucleotide variations (*Life* 2021; 11(7): 633). In the presence of heteroplasmy, there is a threshold level of mutation that is important for both the clinical expression of the disease and for biochemical defects. Many studies have shown that the mutated form is functionally recessive and that a biochemical phenotype is associated with high levels of mutation above a crucial threshold. (*Nat Rev Genet.* 2005; 6(5): 389–402.).

According to the features of mtDNA heterogeneity, there are some challenges to be addressed for mtDNA mutation detection. Firstly, since the acquired mutations are mainly single-nucleotide variation, a highly specific recognition probe is needed. Then, the analysis of whether mtDNA mutation reaches threshold level is important, which challenges the quantitative ability of the detection probes. Finally, since mtDNA heterogeneity develops over time, it is necessary to develop probes suitable for the dynamic monitoring of disease-associated mutations at real time *in vivo*.

According to your suggestion, the following text has been amended to the **Introduction** section (Page 3) in the revised manuscript with turquoise background: “mtDNA mutations are mainly caused by the high level of reactive oxygen species (ROS) generated during oxidative phosphorylation, and the lower-fidelity of DNA replication in mitochondria³.” and “Since mtDNA heterogeneity develops over time, it is necessary to develop probes for monitoring disease-associated mtDNA mutations *in vivo*, which, to the best of our knowledge, has not been developed.”

3 Taylor, R. W. & Turnbull, D. M. Mitochondrial DNA mutations in human disease. *Nat Rev Genet* 6, 389-402, (2005).

have been added to **References** (Page 31) in the revised manuscript as marked with a gray background.

Comment 2:

How stable InCasor is when (i) it encounters the target DNA and (ii) when there is no target DNA, *in-vitro*?

Response 2:

Thank you for the helpful comments. Based on the reviewers' suggestions, we

firstly tested the change of InCasor when it encounters target DNA using dynamic light scattering (DLS) to analyze the particle size changes. As shown in **Figure R2a**, after incubation with 50 nM target dsDNA *in vitro*, the particle size of InCasor reduced from ~300 nm to ~50 nm. Transmission electron microscopy (TEM) imaging was also performed to analyze the morphological change of InCasor with and without target DNA, and an obvious self-degradation of InCasor was observed (**Fig. 1g**).

Subsequently, we examined the stability of InCasor without target DNA. Specifically, we stored InCasor at 4°C for 7 consecutive days to monitor its particle size, zeta potential and ability to generate fluorescent signal in response to the target DNA. The results are shown as followed:

Figure R2. a. DLS analysis of particle size changes of InCasor before and after recognition of target mtDNA *in vitro*. n=3, data show mean \pm SD. **b.** Variation of particle size and potential of InCasor with time at 4°C storage conditions. **c.** The ability of InCasor to recognize target DNA to produce a fluorescent signal over time under storage conditions at 4°C. n=3, data show mean \pm SD.

As the results showed, the particle size of InCasor exhibited a slight decrease at day 7 when stored at 4°C. Notably, storing InCasor at 4°C for 7 days did not significantly affect its capability to generate fluorescent signal in response to target DNA. In summary, InCasor can maintain stability for at least 7 days under appropriate storage conditions (4°C).

According to your suggestion, we revised as followed:

“Transmission electron microscopy (TEM) imaging and dynamic light scattering (DLS) was performed to analyze the morphology change of InCasor before and after incubation with 50 nM target dsDNA. An obvious self-degradation of InCasor was observed (**Fig. 1g** and **Supplementary Fig. 1d**), demonstrating the design principle.”

and “Considering the storage conditions of Cas12a, we further evaluated the long-term stability of InCasor stored at 4°C. As shown in **Supplementary Fig. 1 j-k**, the hydrodynamic dimensions, zeta potential, and mtDNA analysis performance of InCasor did not change notably within one week.” was added to the **Results** section (Page 6-7) as marked with turquoise background. Figure R2 has been amended as **Supplementary Fig. 1 d, j-k** in the revised **Supplementary Information** (Page 11) as marked with turquoise background.

Comment 3:

Characterization of InCasor *in-vivo* is missing. How stable it is *in-vivo* and how it is expelled from the circulatory system?

Response 3:

Thanks for your instructive suggestion. Based on the reviewers' suggestions, we evaluated the *in vivo* behavior of InCasor. First of all, we calculated the half-life of InCasor by detecting the intensity of Cy5 fluorescence signal in blood at different time points after tail vein administration. The mice were intravenously injected with Cy5-ssDNA and Cy5-InCasor (equivalent concentration: 10 nmoles, 200 μ L), respectively. 20 μ L blood was collected from the tail vein at 0.5, 1, 2, 4, 6, 8, 10, 12, and 24 h (n = 3 at each time point), respectively. The samples were diluted with 80 μ L PBS and the fluorescence intensity of Cy5 was measured by Synergy H1 Hybrid Reader (λ_{ex} : 638 nm; λ_{em} : 680 nm). As shown in **Figure R3a**, the circulation half-lives of ssDNA and InCasor in the second ($t_{1/2}(\beta)$) phases were calculated to be 1.33 and 2.63 h, respectively. This might due to the nano size of the preparation enhanced the blood circulation time. The increased circulation time of InCasor in the blood was favorable for its effective tumor accumulation.

The *in vivo* fluorescence imaging and *ex vivo* organ fluorescence imaging was then performed to analyze the *in vivo* distribution and tumor targeting ability of InCasor. After 14 days tumor implantation, mice tail-vein injected with 10 nmoles, 200 μ L Cy5-labelled ssDNA, InCasor_{Sgc8-}, InCasor_{Sgc8+} respectively. The whole-body images were obtained at 2, 4, 6, 8, 12, and 24 h using AniView100 by setting wavelength at

excitation of 640 nm and emission of 710 nm. After tail vein administration for varying periods of time, mice were euthanized with CO₂ and the organs (tumor, heart, lung, liver, spleen, kidney) were removed and *ex vivo* imaged of Cy5 signal using AniView100. The section slides of the tumor, and kidney tissue were imaged by CLSM. The results are shown as followed:

Figure R3. Analysis of the *in vivo* biodistribution of InCasor. **a.** Blood circulation curve of ssDNA and InCasor by measuring the fluorescence intensity of Cy5 in blood at different time points post-injection (n=3, data show mean ± SD.). **b.** *In vivo* biodistribution of Cy5-labelled ssDNA, InCasor_{Sgcb+}, and InCasor_{Sgcb-} in HepG2 tumor-bearing mice over 24 h. **c.** Semi-quantitative analysis of fluorescence intensity of harvested tumor and major organs at 24 h from panel b. **d.** *Ex vivo* imaging of major organs harvested from HepG2 tumor-bearing mice at 2, 4, 6, 8, 12, and 24 h post injection of Cy5-ssDNA, Cy5- InCasor_{Sgcb+}, and Cy5- InCasor_{Sgcb-}. **e.** Semi-quantitative analysis of d. n=3, data show mean ± SD. **f.** Representative images of the tumor, liver, and kidney tissue section in d. Scale bar: 50 μm.

As revealed in **Figure R3 b-c**, after tail vein injection, InCasors_{Sgc8+} firstly accumulated in the tumors and kidneys of HepG2 tumor-bearing mice, and a visible signal appeared in the liver at 6 h post injection. Conversely, Cy5-ssDNA or InCasors_{Sgc8-} preferentially accumulated in the kidney and liver, and their signals reached a maximum value at 2 h post injection. As previously reported, the compact structures together with the negative charges of InCasor would facilitate their glomerular endothelial fenestrae filtration (*J Am Chem Soc.* 2022; 144(51):23522-23533.; *Mol Pharm.* 2008; 5(4):505-15. *Nat Rev Drug Discov.* 2020; 19(10):673-694.).

For further investigation, the mice were sacrificed to harvest the major organs including tumor, heart, lung, spleen, liver, and kidney at 2, 4, 6, 8, 12, and 24 h post injection, followed by *ex vivo* imaging. The *ex vivo* imaging results (**Figure R3c**) and corresponding quantification of fluorescence intensity (**Figure R3d**) were consistent with the results in **Figure R3b**. Notably, both ssDNA and InCasor_{Sgc8-} were able to aggregate in the kidney rather than tumor tissue after injection and cleared within 24 h. In contrast, fluorescence of InCasors_{Sgc8+} was still detected in the tumor 24 h after injection, with a fluorescence signal 1.5-fold stronger than that of InCasor_{Sgc8-}. The results of Cy5 signal in tumor sections further demonstrated the tumor-targeting and retention ability of InCasors_{Sgc8+} (**Figure R3f**). This prolonged retention of InCasors_{Sgc8+} in the tumor facilitates subsequent sensitive monitoring of SNVs in mtDNA.

The Cy5 signal in the kidney sections (**Figure R4**) was recorded by confocal laser scanning microscopy, indicating that the InCasors_{Sgc8+} initially accumulated in the glomerulus, then accumulated in the renal tubules lumina, and gradually passed through the glomeruli to be excreted from the kidney. Collectively, these data demonstrated the preferential tumor accumulation of the InCasor in HepG2 tumor-bearing mice and would be expelled from the kidneys.

Figure R4. Confocal images of kidney sections after intravenous injection of InCasor_{Sgc8-}. Nuclei were stained with DAPI. White dashed circles denote glomeruli and white arrows denote renal tubular lumina. Scale bar: 50 μm .

The *in vivo* stability of InCasor is important for its signal-to-noise ratio in sensing SNV in mtDNA *in vivo*. Therefore, we evaluated the nonspecific signaling generated by InCasor over time in HepG2 tumor-bearing mice using small-animal *in vivo* imaging. Specifically, mice tail-vein injected with 200 μL CLR, DNF/CLR, InCasor_{SNT}, and InCasor_{Cy5} (equivalent amount of CLR: 10 nmoles), respectively. The whole-body images were obtained at 2, 4, 6, 8, 12, and 24 h using AniView100 by setting wavelength at excitation of 640 nm and emission of 710 nm. After 24 h of tail vein administration, mice were euthanized with CO₂ and the organs (heart, lung, liver, spleen, kidney) were removed and *ex vivo* imaged of Cy5 signal using AniView100. The section slides of the liver and kidney tissue were imaged by CLSM. The results are shown as followed:

Figure R5. Analysis of the *in vivo* stability of InCasor. **a.** *In vivo* fluorescence signals of CLR, DNF/CLR, InCasor_{SNT}, and InCasor_{Cy5} in HepG2 tumor-bearing mice over 24 h. n=3 biological repeats. **b.** Fluorescence imaging of harvested tumor and major organs (heart, liver, spleen, lung, and kidney) after 24 h post tail-vein injection of different probe. **c.** Semi-quantitative analysis of b. n=3, data show mean \pm SD. **d.** Representative images of the tumor, liver, and kidney tissue section in d. Scale bar: 50 μm .

As shown in **Figure R5a**, in the free CLR group, a non-specific signal was detected at around 2 h and gradually increased with time, whereas DNF/CLR did not produce

background signal for 24 h, suggesting that the dense structure of DNF protects the CLR from nuclease damage. InCasor_{NT}, which does not target mtDNA, also consistently maintained a lower Cy5 fluorescence signal compared with the CLR group and the InCasor_{Cy5}-positive control group overtime. The *ex vivo* imaging results at 24 h post injection (**Figure R5b**) and corresponding quantification of fluorescence intensity (**Figure R5c**) were consistent with the results in **Figure R5a**. Notably, neither DNF/CLR nor InCasor_{NT} produced non-specific signals in major organs after 24 h injection. As revealed in **Figure R5d**, despite aggregating mainly in the kidney and liver, InCasor_{NT} did not produce non-specific Cy5 fluorescent signals within 24 h due to DNF protection. Collectively, these results indicate that InCasor holds outstanding stability during blood circulation, which reduces the occurrence of false positive signals and lays the foundation for its accurate sensing of SNVs in mtDNA *in vivo*.

According to your suggestion, the following text has been added to the **Results** section in the revised manuscript (Page 24) to clarify *in vivo* behavior of InCasor.:
“Benefiting from the nano meter size, the circulating half-life of InCasor in the second phase ($t_{1/2}(\beta)$) was calculated to be 2.63 h, which was significantly higher than that of ssDNA (1.33 h) (**Supplementary Fig. 15a**). With Sgc8 aptamer, InCasor was able to specifically target tumor tissue in live mice (**Supplementary Fig. 15 b-e**). After circulation, most of InCasor would be expelled from the circulatory system through kidney (**Supplementary Fig. 15f**).

The *in vivo* fluorescence imaging and *ex vivo* organ fluorescence imaging was also performed to analyze the *in vivo* stability of InCasor (**Supplementary Fig. 16**). Due to the integrated nanostructure design, InCasor showed outstanding stability during blood circulation, which reduces the occurrence of false positive signals and lays the foundation for its accurate sensing of SNVs in mtDNA *in vivo*.” And Figure R3 has been amended as **Supplementary Fig. 15**, Figure R5 has been amended as **Supplementary Fig. 16** in the revised **Supplementary Information** (Page 32-33).

“To calculated the half-life of InCasor, the mice were intravenously injected with Cy5-ssDNA and InCasor (equivalent concentration: 10 nmoles, 200 μ L), respectively. 20 μ L blood was collected from the tail vein at 0.5, 1, 2, 4, 6, 8, 10, 12, and 24 h (n = 3

at each time point), respectively. The samples were diluted with 80 μ L PBS and the fluorescence intensity of Cy5 was measured by Synergy H1 Hybrid Reader (λ_{ex} : 638 nm; λ_{em} : 680 nm)” was added to the **Methods** section (Page 47) in the revised manuscript as marked with turquoise background.

Comment 4:

As per my understanding PTK7 is NOT expressed on all adult cells, kindly explain the rationale and provide suitable references.

Response 4:

Thank you for your comment. Protein tyrosine kinase-7 (PTK7), a pseudokinase also known as colon carcinoma kinase (CCK4), was originally identified as a protein overexpressed in colon cancer cell lines (*Oncogene 11(10): 2179-2184, 1995.*). PTK7 is involved in the WNT (named after the Drosophila Wingless (Wg) and the mouse Int-1 genes)-pathways, which play important roles in epithelial mesenchymal transition (EMT) in cancer (*Am J Pathol 2010; 176(6): 2911-2920.*). Although not expressed in all adult cells, PTK7 is upregulated in a variety of malignancies including gastric cancer, lung cancer and acute myeloid leukemia (*Sci. Transl. Med. 2017; 9, eaag2611*). Information regarding PTK7 expression across diverse cell lines originating from different human tissues can be queried in THE HUMAN PROTEIN ATLAS database (<https://www.proteinatlas.org/>), an open access resource for human proteins.

As one of the most widely studied aptamers, Sgc8c has been used to achieve precise delivery of various drugs by specifically binding PTK7 (*ACS Nano 2019, 13, 5852–5863; Angew Chem Int Ed Engl. 2020; 59(5):1897-1905.; J Am Chem Soc. 2023;145(14):7677-7691.*). To confirm the suitability of Sgc8c as a target for the InCasor probe, we identify the expression of PTK7 in HepG2 and MDA-MB-231 cell lines (<https://www.proteinatlas.org/ENSG00000112655-PTK7/cell+line>). The result is shown as followed:

Figure R6. PTK7 expression in human-derived liver cancer (a) and breast cancer cell lines (b).

As shown in **Figure R6**, PTK7 was expressed in both HepG2 and MDA-MB-231 cell lines. Besides, benefitting from the programmability of the RCA template, we can select the appropriate aptamer to construct InCAsor for precise targeting of different target cells.

Comment 5:

Since Mg²⁺ is involved in regulation of ion channels, binds to ATP and affects mitochondria, how does authors ensure that excessive Mg²⁺ has no long term effects on the cells. The authors state that

Response 5:

Thank you for your instructive comment. Magnesium ion (Mg²⁺) are the most abundant intracellular divalent cations, regulated by an array of ion channels to maintain cellular homeostasis (*Nat Commun.* 2023; 14(1):4713.; *Arch. Biochem. Biophys.* 2011; 512, 1–23; *Annu. Rev. Genet.* 2013; 47, 625–646.). Based on the reviewers' recommendations, we have examined the changes in cellular Mg²⁺ levels after InCAsor treatment and its long-term effects on cells. Specifically, Mg-Fluo-4 AM and ICP-MS were used to analyze intracellular Mg²⁺ levels in HepG2 cells treated with

50 nM InCasor for different times (0, 2, 4, 6, 8, 10, 12, 24, and 36 h), respectively. After 8 h of incubation, the InCasor-containing medium was replaced with fresh medium for longer time culture. 50 μ M Mg-Fluo-4 AM was applied to the cells for 20 min, and the cells were visualized by CLSM after staining with Hoechst 33342. The cells were then collected for ICP-MS analysis using the Agilent 7800. Moreover, cell cloning assay was used to analyze the long-term cell viability of HepG2 cells after treated by InCasor probe. The results are shown as followed:

Figure R7. Long-term effects of InCasor on intracellular Mg²⁺ levels and the cytotoxicity. **a.** Schematic and results of using confocal imaging to analyze Mg²⁺ concentration in HepG2 cells treated with 50 nM InCasor for different times. Scale bar: 10 μ m. **b.** Semi-quantitative statistics of Mg-Fluo-4 AM in HepG2 cells after different treatments. n=5, data show mean \pm SD. **c.** ICP-MS was used to analyze Mg²⁺ concentration in HepG2 cells treated with 50 nM InCasor for different times. n=3, data show mean \pm SD. **d-e.** Cell cloning experiment was used to analyze cell viability of HepG2 cells treated with 50 nM DNF/CLR, InCasor_{NT} and InCasor_{ND4}. Data are expressed as mean \pm SD (n = 3 biological repeats).

As shown in **Figure R7 a-b**, the fluorescence intensity of Mg-Fluo-4 AM increased with incubation time, reaching a maximum at 8 h, and then gradually weakened, returning to the original level of the cells at 36 h. This suggested that with the extension of time, the cell's inherent Mg²⁺ regulatory mechanism would expel the large amount of Mg²⁺ delivered by InCasor thereby maintaining intracellular Mg²⁺ homeostasis. The result of ICP-MS further demonstrated that increased intracellular

Mg²⁺ levels due to InCasor uptake returned to steady-state levels with time after cessation of incubation (**Figure R7c**). According to the result of cell cloning experiments (**Figure R7 d-e**), DNF/CLR and InCasor^{NT} did not have significant long-term cytotoxicity, whereas the higher cytotoxicity in the InCasor^{ND4} group may be due to the gene editing. These results indicated that the level of intracellular Mg²⁺ released by InCasor can be regulated over time and have no obvious long-term effects on cells.

According to your suggestion, the following text has been added to the **Results** section in the revised manuscript (**Page 11 and 14**) to clarify the changes in cellular Mg²⁺ levels after InCasor treatment and its long-term effects on cells: “ICP-MS was firstly used to characterize the intracellular Mg²⁺ concentration of InCasor-treated HepG2 cells, which showed around 2-fold increase than untreated control (**Supplementary Fig. 4a**). It’s also found that, the alteration of Mg²⁺ by DNA nanocarrier didn’t significantly change the cell viability after 6 h treatment (**Supplementary Fig. 4b**). To analyze the changes of cellular Mg²⁺ levels after InCasor treatment and its long-term effects on cells, Mg-Fluo-4 AM probe and ICP-MS were used to characterize the intracellular Mg²⁺ levels over time. The result suggested that a large amount of Mg²⁺ can be delivered into cells by InCasor. After removing the InCasor treatment, the intracellular Mg²⁺ levels would decrease over time due to the intrinsic cellular Mg²⁺ regulatory mechanism, thereby maintaining intracellular Mg²⁺ homeostasis (**Supplementary Fig. 4 c-f**). Since Mg²⁺ is one of the most abundant and essential divalent cations among eukaryotic cells, with an intracellular concentration range of 10-30 mM (95% is bound with ATP and other molecules, remaining 0.5-1.2 mM unbound free)³², Mg²⁺ concentration can be regulated by various intracellular mechanism³³.” and “CCK8 experimental and cell cloning experiment results showed that InCasor^{ND4} probe slightly affected cell viability, possibly due to the mitochondrial homeostasis disruption and *ND4* gene silencing (**Supplementary Fig. 7 c-d**).” And Figure R7 has been amended as **Supplementary Fig. 4 c-f** and **Supplementary Fig. 6 c-d** in the revised **Supplementary Information** (**Page 16 and 18**).

“To analyze the change of intracellular Mg²⁺ levels in a long time, HepG2 cells were treated with 50 nM InCasor for different times (0, 2, 4, 6, 8, 10, 12, 24, and 36 h),

respectively. After 8 h of incubation, the InCasor-containing medium was replaced with fresh medium for longer time culture. 50 μ M Mg-Fluo-4 AM was applied to the cells for 20 min, and the cells were visualized by CLSM after staining with Hoechst 33342. The cells were then collected for ICP-MS analysis using the Agilent 7800.” was added to the **Methods** section (Page 42) in the revised manuscript as marked with turquoise background.

33 Li, M. *et al.* Molecular basis of Mg²⁺ permeation through the human mitochondrial Mrs2 channel. *Nat Commun* **14**, 4713, (2023).
have been added to **References** (Page 32) in the revised manuscript as marked with a gray background.

Comment 6:

Since Mg²⁺ is already in high concentrations inside the cells and mitochondria, did authors tested InCasor activity with providing Mg²⁺ externally. Please provide data in relation to cellular Mg²⁺.

Response 6:

Thank you for the helpful comments. Although the total intracellular Mg²⁺ content in most celltypes is to 10 ~30 mM, the free Mg²⁺ concentration is typically in the range of 0.5– 1.2 mM and a considerable proportion of the complexed Mg²⁺ pool is bound as Mg²⁺-ATP (*Physiol. Rev.* 2015; 95, 1–46.; *Cell* 2020;183, 474–489 e17.; *Arch. Biochem. Biophys.* 2011; 512, 1–23.). According to the effect of *in vitro* Mg²⁺ concentration on the trans-cleavage activity of Cas12a (**Fig. 1b**), we hypothesized that the original intracellular level of free Mg²⁺ could not provide an ideal ionic environment for Cas12a. To demonstrate the importance of providing additional Mg²⁺ in activating the trans-cleavage of Cas12a/crRNA in live cells, we prepared another InCasor^{ND4}-Co²⁺ probe by replacing the encapsulated Mg²⁺ with Co²⁺.

The preparation of DNF-Co²⁺ hybrid nanoflowers were as follows. Template_{MUC1} was cyclized using CircLigase II (Lucigen) according to the manufacturer’s instructions. In short, 5 μ L 10 mM 5' phosphorylated Template_{MUC-1} and 1 μ L CircLigase II ssDNA Ligase (5 U/ μ L) in 50 μ L of 1 \times CircLigase II reaction buffer (33 mM Tris-acetate (pH

7.5), 2.5 mM CoCl₂, 66 mM potassium acetate, and 0.5 mM DTT) was incubated at 60°C for 16 h before heating to 80°C for 10 minutes to denature the enzyme. A solution containing 50 μL 1 mM cyclized Template_{MUC-1}, 7.5 μL 10 mM Primer_{MUC-1}, 1× RCA reaction buffer (50 mM Tris buffer (pH 7.0), 10 mM CoCl₂, 10 mM (NH₄)₂SO₄) was heated to 95°C and cooled to room temperature at a rate of 0.1°C/s. After that, 20 μL 10 mM dNTPs and 5 μL 10000 U/mL Phi 29 DNA polymerase were added and the mixture incubated at 30°C for 20 h. The sample was then heated to 65°C for 10 minutes and cooled to room temperature. The resulting precipitate was washed 5 times by resuspending in DEPC-treated H₂O after collected by centrifugation. The preparation of InCasor_{ND4}-Co²⁺ was similarly to InCasor_{ND4}-Mg²⁺. Then the InCasor_{ND4}-Co²⁺ were analyzed by agarose gel electrophoresis, DLS and SEM-based EDS. In addition, the uptake of InCasor_{ND4}-Co²⁺ by HepG2 cells was examined by CLSM. The results are shown as followed:

Figure R8. Preparation and characterization of the InCasor_{ND4}-Co²⁺ probe. a. Synthesis of InCasor-Co²⁺ probe by replacing Mg²⁺ with Co²⁺ in rolling circle amplification reaction. **b.** Agarose gel electrophoresis analysis of synthesized InCasor-Mg²⁺ and InCasor-Co²⁺ probe: Lane 1: DNA ladder, Lane 2: Template, Lane 3: Primer, Lane 4: Ligation, Lane 5: InCasor-Mg²⁺, Lane 6: InCasor-Co²⁺. **c.** Particle sizes and Zeta potential of InCasor-Mg²⁺ and InCasor-Co²⁺ probe. n=3, data show mean ± SD. **d.** Atomic fraction analysis of the InCasor-Mg²⁺ and InCasor-Co²⁺. n=3, data show mean ± SD. **e.** Confocal imaging of HepG2 cells incubated with 50 nM FAM-labelled InCasor_{ND4}-Mg²⁺ or InCasor_{ND4}-Co²⁺ probe. Scale bar: 10 μm.

As shown in **Figure R8 a-c**, InCasor_{ND4}-Co²⁺ showed a nanoflower structure with

a diameter of 282.7 ± 25.2 nm and a potential of -22.5 ± 0.79 mV, which was similar to InCasor_{ND4}-Mg²⁺. The results of atomic fraction analysis showed that the InCasor_{ND4}-Co²⁺ is enriched in Co²⁺ but not Mg²⁺ (**Figure R8d**). Moreover, InCasor_{ND4}-Co²⁺ can be efficiently taken up by HepG2 cells, which was similar to InCasor_{ND4}-Mg²⁺.

We then examined the ability of InCasor_{ND4}-Co²⁺ to sense the target mtDNA in living cells. qPCR was firstly used to analyze the gene editing efficiency of InCasor_{ND4}-Co²⁺. Then, CLSM and flow cytometry were used to analyze the fluorescence signal generated by InCasor_{ND4}-Co²⁺ in HepG2 cells. The results are shown as followed:

Figure R9. The ability of InCasor_{ND4}-Co²⁺ probe to recognize and sense target mtDNA in living cells. a. Analysis of *ND4* expression in HepG2 cells treated with InCasor_{ND4}-Mg²⁺ or InCasor_{ND4}-Co²⁺ using qPCR. n = 3 biological repeats, data show mean \pm SD. **b-c.** Confocal imaging (left) and flow cytometry analysis (right) of HepG2 cells treated with InCasor_{ND4}-Mg²⁺ or InCasor_{ND4}-Co²⁺. Scale bar: 10 μ m.

As the results showed, both probes showed efficient mtDNA gene silencing ability, while the InCasor_{ND4}-Co²⁺ probe did not produce obvious fluorescence signals in live HepG2 cells, indicating that a sufficient Mg²⁺ supply is crucial for activation of the trans-cleavage activity of Cas12a/crRNA in live cells.

To further examine the necessity of providing additional Mg²⁺, we treated cells with DPBS containing different concentrations of Mg²⁺ and examined intracellular Mg²⁺ levels and the mtDNA sensing ability of InCasor_{ND4}-Co²⁺ using CLSM. Specifically, cells were plated one day before experiment and the medium was subsequently replaced with DPBS containing different concentrations of Mg²⁺ (0, 1, 3, 5, 7, 10, 15, and 20 mM) after cultured overnight. After 12 h of incubation, 50 μ M Mg-Fluo-4 AM was applied to the cells for 20 min. And the cells were visualized by CLSM after staining with Hoechst 33342. Then, the cultured cells were incubated in DPBS

containing different concentrations of Mg^{2+} (0, 1, 3, 5, 7, 10, 15, and 20 mM) and 50 nM InCasor_{ND4}-Co²⁺. Mito-Tracker Red and Hoechst 33342 were used to stain mitochondria and cell nucleus respectively before imaging. The results are shown as followed:

Figure R10. a. Confocal microimaging was used to analyze the fluorescence signal of Mg-Fluo-4 AM in HepG2 cells cultured in PBS with different concentrations of Mg^{2+} for 12 h. Scale bar: 10 μ m. **b.** Semi-quantitative statistics of Mg-Fluo-4 AM in HepG2 cells after different treatments. $n=5$ biological repeats, data show mean \pm SD. **c.** Confocal imaging of HepG2 cells treated with InCasor_{ND4}-Co²⁺ in DPBS containing different concentrations of Mg^{2+} . Scale bar: 10 μ m. **d.** Semi-quantitative analysis of fluorescent intensity in panel c is shown. $n=5$, data show mean \pm SD. **e.** Correlation analysis of the fluorescence intensity of InCasor_{ND4}-Co²⁺ and the fluorescence intensity of Mg-Fluo-4 AM. $n=5$ biological repeats, data show mean \pm SD.

As the results showed, the intensity of Mg-Fluo-4 AM in HepG2 cells increased with the concentration of Mg^{2+} in DPBS (**Figure R10 a-b**). Notably, the intensity of Cy3, which was generated by InCasor_{ND4}-Co²⁺, also increased with the concentration of Mg^{2+} in DPBS (**Figure R10 c-d**). The correlation analysis of Cy3 intensity and Mg-Fluo-4 AM showed that the sensing ability of InCasor_{ND4}-Co²⁺ was positively correlated with the intracellular Mg^{2+} level. Collectively, these data suggest that increasing intracellular Mg^{2+} levels contributed to the intracellular activation of Cas12a

trans-cleavage activity, which is one of the most important functions of InCasor.

According to your suggestion, we revised as followed:

“To demonstrate the importance of providing additional Mg^{2+} in activating the trans-cleavage of Cas12a/crRNA in live cells, we prepared another InCasor_{ND4}- Co^{2+} probe by replacing the encapsulated Mg^{2+} with Co^{2+} (**Supplementary Fig. 10**). Although both probes showed efficient mitochondrial targeting and mtDNA gene silencing ability (**Fig. 3d**), the InCasor_{ND4}- Co^{2+} probe did not produce obvious fluorescence signals in live HepG2 cells (**Fig. 3 e-f**), indicating that a sufficient Mg^{2+} supply is crucial for activation of the trans-cleavage activity of Cas12a/crRNA in live cells. To further examine the necessity of providing additional Mg^{2+} , the cells treated with InCasor_{ND4}- Co^{2+} probe were incubated with DPBS containing different concentrations of Mg^{2+} . The intracellular Mg^{2+} levels and the mtDNA sensing ability of InCasor_{ND4}- Co^{2+} were then examined using CLSM. As shown in the results, the intensity of Mg-Fluo-4 AM and the intensity of Cy3 in HepG2 cells both increased with the concentration of Mg^{2+} in DPBS (**Fig. 3 g-j**). The correlation analysis of Cy3 intensity and Mg-Fluo-4 AM showed that the sensing ability of InCasor_{ND4}- Co^{2+} was positively correlated with the intracellular Mg^{2+} level (**Fig. 3k**). Collectively, these data suggest that increasing intracellular Mg^{2+} levels contributed to the intracellular activation of Cas12a trans-cleavage activity.” was added to the **Results** section (Page 14-15) as marked with turquoise background. Figure R8 has been amended as **Supplementary Fig. 10**, and Figure R9-10 has been amended as **Fig. 3 d-k**, in the revised **Supplementary Information** (Page 24) and the revised **Results** section (Page 16) marked with turquoise background.

“Preparation and characterization of DNA- Co^{2+} hybrid nanoflower.

A new DNF template encoding MUC1 aptamer was designed to synthesize DNF_{MUC1}- Mg^{2+} and DNF_{MUC1}- Co^{2+} . DNF_{MUC1}- Mg^{2+} were synthesized by above-described method. The preparation and characterization of DNF- Co^{2+} hybrid nanoflowers were as follows. Template_{MUC1} was cyclized using CircLigase II (Lucigen) according to the manufacturer’s instructions. In short, 5 μ L 10 mM 5' phosphorylated

Template_{MUC-1} and 1 μ L CircLigase II ssDNA Ligase (5 U/ μ L) in 50 μ L of 1 \times CircLigase II reaction buffer (33 mM Tris-acetate (pH 7.5), 2.5 mM CoCl₂, 66 mM potassium acetate, and 0.5 mM DTT) was incubated at 60°C for 16 h before heating to 80°C for 10 minutes to denature the enzyme. A solution containing 50 μ L 1 mM cyclized Template_{MUC-1}, 7.5 μ L 10 mM Primer_{MUC-1}, 1 \times RCA reaction buffer (50 mM Tris buffer (pH 7.0), 10 mM CoCl₂, 10 mM (NH₄)₂SO₄) was heated to 95°C and cooled to room temperature at a rate of 0.1°C/s. After that, 20 μ L 10 mM dNTPs and 5 μ L 10000 U/mL Phi 29 DNA polymerase were added and the mixture incubated at 30°C for 20 h. The sample was then heated to 65°C for 10 minutes and cooled to room temperature. The resulting precipitate was washed 5 times by resuspending in DEPC-treated H₂O after collected by centrifugation. Then the DNFs were analyzed by agarose gel electrophoresis, DLS and SEM-based EDS.” was added to the **Methods** section (Page 36) in the revised manuscript as marked with turquoise background.

Comment 7:

What is the sensitivity of the mutation detection system, Can it identify 1 mutations in one of the mitochondria? What is the estimated minimum number of mutations required? Can the signal be quantified?

Response 7:

Thank you for the helpful comments. Following the reviewer's suggestion, we firstly evaluated the sensitivity of InCasor by *in vitro* determination of the limit of detection. Specifically, 25 μ L of 1 μ M InCasor, 5 μ L dsDNA with different concentration and 10 μ L of 10 \times Tolo buffer and 70 μ L DEPC-treated H₂O were mixed. After incubated at 37°C for 30 minutes, the fluorescence of each reaction mixture was measured (λ_{ex} : 488 nm; λ_{em} : 520 nm). Further, we tested the effect of free Cas12a/crRNA complex and different fluorescent reporter motifs (single-stranded reporter (SSR), circular reporter (CLR), DNF/SSR, and DNF/CLR) on the detection sensitivity. The results are shown as followed:

Figure R11. Comparison of the mtDNA detection efficiency of the SSR (a), CLR (b), DNF/SSR (c), DNF/CLR (d), and InCasor (e). $n=3$, data show mean \pm SD.

As shown in **Figure R11**, the limit of detection of InCasor for dsDNA_{ND4} is 1 fM, considerably lower than the other control groups (Cas12a/crRNA + SSR: 10 pM; Cas12a/crRNA + CLR: 1 pM; Cas12a/crRNA + DNF/SSR: 100 fM; Cas12a/crRNA + DNF/CLR: 10 fM). The increased sensitivity of InCasor may attribute to several factors. Firstly, the CLR exhibited a more favorable response to the trans-cleavage activity of Cas12a compared to SSR; Secondly, the integration of the nanoprobe led to an increased local concentration of CLR, facilitating proximity between Cas12a/crRNA and CLR and this proximity enhancement improved reaction kinetics.

Mitochondria are organelles characterized by their oval shape, typically 0.5-1.0 μm in diameter and 1.5-3.0 μm in length (*Nature. 2021; 593(7859): 435-439.*). To determine the concentration when single mitochondrion contains one copy of mutant mtDNA, we approximated the mitochondrion's shape as a spheroid with a diameter of approximately 1 μm . Thereby, the theoretical concentration of 1 mtDNA in 1 mitochondria is calculated to be 3.19 fM. According to the detection limit of InCasor for dsDNA_{ND4} (1 fM), theoretically, InCasor can detect one mutant mtDNA within a single mitochondria.

Can the signal be quantified?

For signal quantification, when performing *in vitro* experiments, the signal of InCasor can be quantified by a fluorescence spectrophotometer. When observing target genes in living cells or *in vivo*, the signal of InCasor cannot be absolutely quantified due to the settings of laser intensity, gain multiplier and exposure time. Nevertheless, a semi-quantitative analysis can be conducted to analyze the signal intensity.

According to your suggestion, the following text has been added to the **Results** section in the revised manuscript (Page 7) to clarify the sensitivity of InCasor: “The limit of detection of InCasor for dsDNA_{ND4} analysis was calculated to be around 1 fM, considerably lower than the other control groups (**Supplementary Fig. 1 e-i**). The increased sensitivity of InCasor may attribute to several factors. Firstly, the CLR exhibited a more favorable response to the trans-cleavage activity of Cas12a compared to single-stranded reporter (SSR); Secondly, the integration of the nanoprobe led to an increased local concentration of CLR, facilitating proximity between Cas12a/crRNA and CLR. This proximity improved reaction kinetics, further contributing to the increase of sensitivity.” And Figure R11 has been amended as **Supplementary Fig. 1 e-i** in the revised **Supplementary Information** (Page 11) as marked with turquoise background.

Comment 8:

Sensitivity of *in-vivo* imaging using fluorochromes has very limited usability.

Response 8:

Thanks for your comment. We agree that the output mode of fluorescence signals limits the usability of InCasor for *in-vivo* imaging. To increase the versatility of InCasor's imaging, some other signal output modalities, such as MRI and positron emission tomography (PET) scanning modalities, may be further investigated (*Nat Mater.* 2021; 20(5):585-592.; *Nature.* 2019; 575(7782): 380-384.; *Nat Biomed Eng.* 2023; 7(3):313-322.).

According to your suggestion, we revised as followed:

“To increase the versatility of InCasor's imaging, some other signal output

modalities, such as MRI and PET scanning modalities^{53,54}, may be further investigated.”

was added to the **Discussion** section (Page 29) as marked with turquoise background.

53 Farhadi, A., Sigmund, F., Westmeyer, G. G. & Shapiro, M. G. Genetically encodable materials for non-invasive biological imaging. *Nat Mater* **20**, 585-592, (2021).

54 Momcilovic, M. *et al.* In vivo imaging of mitochondrial membrane potential in non-small-cell lung cancer. *Nature* **575**, 380-384, (2019).

have been added to **References** (Page 33) in the revised manuscript as marked with a gray background.

Comment 9:

Please show self degradation/half life on InCasor inside the mitochondria.

Response 9:

Thank you for the helpful comments. Based on the reviewer's suggestion, we used CLSM to analyze the self-degradation process of InCasor inside the mitochondria. The InCasor probe used in this experiment consisted of FITC-DNF, Cy5 and BHQ2 labelled CLR, and Cas12a/crRNA_{ND4} complex, named FITC-InCasor_{ND4}. We prepared FITC-DNF by performing the RCA reaction using amino-modified dUTP and modifying FITC-NHS to DNF by the EDC-NHS reaction. Then, HepG2 cells were incubated with FITC-InCasor_{ND4} or FITC-InCasor_{NT} respectively for different time. After 6 h of incubation, fresh medium was used to replace the medium containing InCasor for longer incubations. Afterwards, the mitochondria were labeled using Mito Tracker Red and nuclei were labeled using Hoechst. Finally, the fluorescence intensity of InCasor's FITC and Cy5 was observed within the mitochondria using laser confocal microscopy. The results are shown as followed:

Figure R12. The self-degradation of InCasor inside the mitochondria. a-b. Confocal imaging was used to analyze the intensity of FITC fluorescence signal and Cy5 fluorescence signal in HepG2 cells changed with time after FAM-InCasor_{ND4} (a) or FITC-InCasor_{NT} (b) treatment. Scale bar: 10 μ m. **c.** Semi-quantitative analysis of fluorescent intensity of FITC in panel a-b is shown. n=5 biological repeats. **d.** Semi-quantitative analysis of fluorescent intensity of Cy5 in panel a-b is shown. n=5 biological repeats, data show mean \pm SD.

As shown in **Figure R12 a-b**, both green fluorescent signals of DNF and red signals generated in response to the target site could be observed in the mitochondria of HepG2 cells in FITC-InCasor_{ND4} group after 6 h of incubation, whereas FITC signal were observed in FITC-InCasor_{NT} group. After changing the medium, the green fluorescence signal of FITC-InCasor_{ND4} began to gradually decrease. The red fluorescence signal was always co-localized with mitochondria and reached its maximum at 8 h, followed by a gradual decrease. In FITC-InCasor_{NT} group, the green fluorescence signal remained stable at 8 h, and then weakened with time. Normalized to a 6 h fluorescent signal, the half-life of FITC-InCasor_{ND4} was calculated to be \sim 9.25 h, the half-life of FITC-InCasor_{NT} was \sim 11.53 h, and the half-life of Cy5 produced in

response was ~11.72 h (**Figure R12 a-b**). The shorter half-life of FITC-InCasor^{ND4} may be due to the activation of Cas12a's trans-cleavage activity by *ND4* gene to cleave DNF, thus accelerating its self-degradation process.

According to your suggestion, the following text has been added to the **Results** section in the revised manuscript (**Page 13**) to clarify the self-degradation on InCasor inside the mitochondria: “The metabolic behavior of InCasor inside mitochondria was then examined. Specifically, a FITC labelled DNF was used to prepare FITC-InCasor^{ND4} and FITC-InCasor^{NT}. As shown in **Supplementary Fig. 6**, both FITC and Cy3 signal could be observed in the mitochondria of HepG2 cells in FITC-InCasor^{ND4} group after 6 h incubation, whereas only FITC signal were observed in FITC-InCasor^{NT} group. The half-life of FITC-InCasor^{ND4} was calculated to be ~9.25 h, the half-life of FITC-InCasor^{NT} was calculated to be ~11.53 h, and the half-life of Cy5 produced in response was calculated to be ~11.72 h. The shorter half-life of InCasor^{ND4} may be due to the activation of Cas12a's trans-cleavage activity to cleave DNF, thus accelerating its self-degradation process.” And Figure R12 has been amended as **Supplementary Fig. 6** in the revised **Supplementary Information** (**Page 19**).

Comment 10:

Please share data regarding non-specific localization of InCasor in cytoplasm and/or nucleus.

Response 10:

Thank you for the comment. The distribution of InCasor in subcellular organelles of living cell were analyzed by laser confocal microscope. To simultaneously observe the subcellular organelle distribution and its nonspecific signaling in the cytoplasm, we prepared FITC-DNF by performing the RCA reaction using amino-modified dUTP and modifying FITC-NHS to DNF by the EDC-NHS reaction. Then, HepG2 cells were incubated with FITC-InCasor^{NT} for 6 h. Afterwards, the mitochondria were labeled using Mito Tracker Red and nuclei were labeled using Hoechst. Finally, fluorescence signals were imaged using laser confocal microscope to analyze the subcellular organelle distribution of InCasor. The results are shown as followed:

Figure R13. The self-degradation of InCasor inside the mitochondria. a. Confocal micrographs showing the subcellular localization and generated fluorescence signal of InCasor_{NT} in HepG2 cells. Cas12a protein on InCasor was labeled with Cy5 (purple), crRNA on InCasor was labeled with FITC (green), and mitochondria were stained with Mito Tracker (red). Scale bar: 25 μm. **b.** The Pearson's correlation coefficient (PCC) of InCasor and mitochondrial or nucleus were statistically analyzed by Fiji. n=5 biological repeats, data show mean ± SD.

We selected five CLSM images of InCasor_{NT} to analyze the non-specific localization of InCasor in cytoplasm and/or nucleus. As the results showed, due to the encoded Cytochrome C aptamer (Cyt C apt), InCasor_{NT} co-localized well with mitochondria, while no obvious cell nucleus co-localization was observed. Although a small portion of InCasor_{NT} was distributed in the cytoplasm, no nonspecific Cy5 signaling was observed. To analyze the correlation between the fluorescence signal of InCasor_{NT} and the nucleus, we calculated the Pearson correlation coefficient (PCC). As shown in **Figure R13b**, the PCC coefficient of InCasor_{NT} with the nucleus was around -0.15, indicating that InCasor was not delivered into the nucleus, which may be due to the limitation of small nuclear pore size and the absence of nucleus-targeting motif (such as NLS).

Response to reviewer 3:

Overview:

The authors propose a novel application of Cas12a trans-cleavage combined with DNA nanoflower for single nucleotide variation detection in live cells. They call this system, “InCasor”. Through a series of controlled assays in hepatocytes, the authors detected efficient cleavage of a circular DNA reporter and uptake of DNFs into mitochondria. They further demonstrate concentration dependency of detection with a threshold of <10nM InCasor, high mtDNA target specificity with essentially no nuclear off-target activity, and compatibility across multiple cell lines. Additionally, the authors were able to engineer the InCasor system to detect SNVs outside canonical PAM sites and showed highly specific detection of mutant mitochondria in a laboratory-generated heteroplasmic cell line. Finally, in a mouse model, injected InCasor was able to specifically localize to a tumor site only in the presence of the Sgc8 aptamer (targeting PTK7 receptor) and engineered crRNAs targeting mutations with one or two mismatches.

General comments:

Overall, this is a well written account of a detailed study, utilizing a wide array of techniques, describing the potential use of Cas12a (InCasor) for the detection of single mutations in mitochondrial DNA that are often associated with human disease states, including metastatic cancers. The individual experiments are well-conceived, controlled, and clearly presented. Collectively, they suggest that the InCasor has potential utility for SNV detection across a wide range of diagnostic applications. In my opinion, the study is highly relevant as a proof-of-concept and would be interesting to the general audience of the journal. I offer the following suggestions for revision for the authors to consider.

Response:

Thanks for your appreciation for our work with instructive suggestions. Based on your suggestions, we have added some new data. And all the modified portions were highlighted in yellow.

Comment 1:

The use of Cas12a as a detection system has expanded greatly over the past few years.

The introduction would have more impact if it were expanded to provide additional context regarding the novelty of the current study, considering the breadth of Cas12a applications.

Response 1:

Thank you for your comment. Cas12a (also called Cpf1) is a type V CRISPR-Cas protein that contains an RuvC endonuclease domain for DNA cleavage, including targeted double-stranded (ds)DNA and trans-cleaved collateral ssDNA (*Science*. 2018; 360(6387):436-439.), which has been exploited in nucleic acid or small molecules detection platforms (*Trends Analyt Chem*. 2023;160:116980. *Trends Biotechnol*. 2022; 40(11):1326-1345.). Researchers have coupled the CRISPR/Cas12a system with nucleic acid amplification strategies, such as nucleic acid sequence-dependent amplification (NASBA), polymerase chain reaction (PCR), recombinase polymerase amplification (RPA), rolling-circle amplification (RCA), and ring-mediated isothermal amplification, for rapid detection of specific pathogens in clinical samples (*Chem Soc Rev*. 2021; 50(21):11844-11869.).

The cis- and trans-cleaving activity of Cas12as have been used to monitor nucleic acids, small molecules, ions and other biomarkers in live cells (*Chem. Sci.*, 2022; 13(15):4364-4371.; *Anal Chem*. 2022; 94(28): 10159-10167.; *Small* 2021, 2104622.). For example, Song et al. introduce an organic framework-sheltering CRISPR/aptamer-based sensor for ATP imaging *in vivo* (*Biosensors and Bioelectronics* 2022; 209, 114239.). Li et al. developed a CRISPR/Cas12a biosensor for *in vivo* spatiotemporally imaging of mRNA (*Biosensors and Bioelectronics* 2022; 216, 114646.). However, to the best of our knowledge, tools for imaging of mtDNA mutations *in vivo* haven't been previously reported. For mtDNA mutation detection, the Cas12a probes need to be efficiently delivered into mitochondria and the SNV recognition specificity of Cas12a needs further improved. In addition, to achieve sensitive detection of mtDNA mutation, the collateral trans-cleavage activity of Cas12a in live cells need to be fully activated.

In this work, we designed an integrated nano Cas12a sensor (InCasor), which was able to efficiently deliver Cas12a into mitochondria in live cells for recognizing target mtDNA, which will activate the trans-cleavage activity of Cas12a to cleave the co-

delivered circular reporter (CLR), generating robust fluorescence signals to report mtDNA mutations (illustrated in **Fig. 1a**). By crRNA engineering, we can specifically identify SNVs in target mtDNA. Besides, by supplying sufficient amount of Mg^{2+} into cellular environment, the collateral trans-cleavage activity of Cas12a can be significantly enhanced, amplifying the detection signal in live cells. We have demonstrated that InCasor is able to identify mtDNA mutations *in vivo*, thereby showing high potential for various biological and biomedical applications.

According to your suggestion, we revised as followed:

“However, most of them used dCas9¹⁴ for imaging of genomic loci dynamics in live cells^{16,18}, while single-nucleotide variation (SNV) imaging remains a challenge¹⁹. To solve this issue, we tend to focus on other subtypes of CRISPR systems.” And “Recently, CRISPR/Cas12a have been used to monitor nucleic acids, small molecules, ions and other biomarkers in live cells and *in vivo*²⁴⁻²⁶. For example, Song et al. introduce an organic framework-sheltering CRISPR/aptamer-based sensor for ATP imaging *in vivo*²⁵. Li et al. developed a CRISPR/Cas12a biosensor for *in vivo* spatiotemporally imaging of mRNA²⁷. However, for mtDNA mutation detection, the Cas12a probes need to be efficiently delivered into mitochondria and the SNV recognition specificity of Cas12a needs further improved. In addition, to achieve sensitive detection of mtDNA mutation, the collateral trans-cleavage activity of Cas12a in live cells need to be fully activated.

In this work, we designed an integrated nano Cas12a sensor (InCasor), which was able to efficiently deliver Cas12a into mitochondria in live cells for recognizing target mtDNA, which will activate the trans-cleavage activity of Cas12a to cleave the co-delivered circular reporter (CLR), generating robust fluorescence signals to report mtDNA mutations (illustrated in **Fig. 1a**). By crRNA engineering, we can specifically identify SNVs in target mtDNA. Besides, by supplying sufficient amount of Mg^{2+} into cellular environment, the collateral trans-cleavage activity of Cas12a can be significantly enhanced, amplifying the detection signal in live cells. We have demonstrated that InCasor is able to identify mtDNA mutations *in vivo*, thereby showing high potential for various biological and biomedical applications.” was added

to the **Introduction** section (Page 4-5) as marked with yellow background.

27 Liu J, *et al.* Smart NIR light-gated CRISPR/Cas12a fluorescent biosensor with boosted biological delivery and trans-cleavage activity for high-performance *in vivo* operation. *Biosens Bioelectron* **216**, 114646, (2022).

have been added to **References** (Page 32) in the revised manuscript as marked with a gray background.

Comment 2:

Most of the experiments and statistical analyses comprise only three replicates. Although this may be acceptable, the replicates also show remarkably similar magnitudes; i.e. little variance, which is somewhat surprising for what are presumed to be biological replicates. The authors should address the nature of the replicates in the Methods and mention this as a potential limitation to their outcomes in the Discussion.

Response 2:

Thank you for your kind suggestion. All replicated experiments in this study were biological replicates with sample sizes ranging from 3, 5, 6, to 50 (for single cell or single mitochondria analysis). The number of independent replicate samples (n) and the relevant statistical parameters for each experiment (such as mean or standard deviation) are described in the pertinent figure legends. No statistical methods were used to pre-determine sample sizes. We have added the detail principle of experiment replicates in the **Methods**.

The majority of biological indicators are not a definite value, but are assumed to have a normal distribution. From a statistical perspective, a single measurement lacks the capacity to provide insights into this distribution. In the pursuit of balancing the interplay between cost and confidence, we performed each experiment with a reasonable repetition to obtain a reasonable approximation of the true distribution while optimizing resource allocation. The experiments with little variance were mostly *in vitro* fluorescence experiments, which is consistent with the literature report about the use of CRISPR systems for *in vitro* nucleic acid detection (*Angew Chem Int Ed Engl.* 2023; 62(32): e202305536.; *Nat Commun.* 2020; 11(1): 4711.; *ACS Nano.* 2023;

17(13): 12903-12914.). All cell and animal experiments were ≥ 3 biological replicates.

According to your suggestion, the following text has been added to the **Methods** section in the revised manuscript (Page 50) to clarify the nature of the repetition:

“Statistics and reproducibility

All the experiments were independently performed at least three times unless otherwise stated. The number of independent replicate samples (n) and the relevant statistical parameters for each experiment (such as mean or standard deviation) are described in the pertinent figure legends. No statistical methods were used to pre-determine sample sizes. GraphPad Prism software was used to perform all statistical analyses. The results were expressed as a mean \pm SD. Significant differences between different groups were determined using two-tailed unpaired Student’s t-tests for two-group comparisons and one-way analysis of variance (ANOVA) with post hoc Tukey’s test for multiple-group comparisons (* $P < 0.05$, ** $P < 0.01$, *** $P < 0.001$ and **** $P < 0.0001$; ns: no significance).”

Comment 3:

The study is also restricted to mitochondrial genes N4 & N5 and a couple of SNVs. This should be included in the Discussion as another possible limitation to broad applicability; e.g. more genes and SNVs should be tested. Would the authors suggest particular genes/SNVs for follow on studies?

Response 3:

Thanks for your suggestion. We agree that the versatility is very important to the significance of InCasor. Therefore, we performed additional experiments to further investigate the broad applicability of InCasor. Specifically, we selected two loci with mutations at position 12 or 16 nt away from PAM, respectively, and modified the crRNA to improve the recognition specificity. mt.4769A>G, located in *ND2* gene, was selected based on the sequencing results of MDA-MB-231 used in this study. mt.3916G>A, located in *ND1* gene, was selected according to the literature (*Hum. Mutat.* 2020, 41, 2028–2057; *Nat Rev Cancer.* 2021, 21(7):431-445.), which was

related to oncocytoma. For each mutation site, we designed five crRNAs separately: fully matched with the mutant site, named crRNA_{FM}; inset single mismatch near PAM, named crRNA_{SM1}; inset single mismatch near SNP site, named crRNA_{SM2}; inset double mismatches near PAM, named crRNA_{DM1}; inset double mismatches near SNP site, named crRNA_{DM2}. After synthesized crRNAs through *in vitro* transcription, an *in vitro* fluorescence analysis method (as described in **Methods**) was employed to distinguish between wild-type and mutant mtDNA. And the discrimination factors (DFs) were calculated. The results are shown below:

Figure R14. Engineering crRNA for recognition of mutations in mtDNA that are distant from the PAM. **a.** Sequences of crRNA_{FM} (fully matched with the 4769A>G mutant *ND2* gene) and engineered crRNAs (with different numbers of nucleotide mismatches at different position). Each SNV is indicated by a bold red letter. Each mismatched position is indicated by an underlined bold blue letter. **b.** An *in vitro* fluorescence assay was used to estimate the ability of different InCasor probes to detect the 4769A>G mutation in the *ND2* gene. n=3, data show mean ± SD. **c.** The discrimination factors of the InCasor probe with different crRNAs toward WT and MT mtDNA at 4769A>G. **d.** Sequences of crRNA_{FM} (fully matched with the 3916G>A mutant *ND1* gene) and engineered crRNAs (with different numbers of nucleotide mismatches at different position). Each SNV is indicated by a bold red letter. Each mismatched position is indicated by an underlined bold blue letter. **e.** An *in vitro* fluorescence assay was used to estimate the ability of different InCasor probes to detect the 3916G>A mutation in the *ND1* gene. **f.** The discrimination factors of the InCasor probe with different crRNAs toward WT and MT mtDNA at 3916G>A. n=3, data show mean ± SD. The sequences of all crRNAs used are listed in Supplementary Table 1.

As shown in **Figure R14**, when the mutation site is distant from the PAM sequence,

the fully matched crRNA and crRNA_{SM1} cannot distinguish between wild-type and mutant mtDNA. In contrast, the insertion of two discrete mismatches in the seed region (crRNA_{DM1}) and the insertion of 1 mismatch in the SNP locus (crRNA_{SM2}) both showed improved DFs (for mt.4769A>G, DF of crRNA_{SM2} was 4.1, DF of crRNA_{DM1} was 3.6; for mt.3916G>A, DF of crRNA_{SM2} was 2.4, DF of crRNA_{DM1} was 2.2). According to the results, engineering the crRNA could improve the ability of InCasor for sensing SNP sites distant from the PAM sequence.

Would the authors suggest particular genes/SNVs for follow on studies?

The particular genes/SNVs for future studies could be focusing on some important mutations in mtDNA that could cause major disease and disability (*Trends Cell Biol.* 2022; 32(5):391-405; *Nat Rev Genet.* 2005; 6(5): 389–402.). For example, the tRNA^{Lys} (MT-TK) m.8344A>G mutation, which causes myoclonic epilepsy and ragged red fibre (MERRF) syndrome. The tRNA^{Leu} (UUR) (MT-TL1) m.3243A>G mutation causes mitochondrial encephalomyopathy, lactic acidosis and stroke-like episodes (MeLAS) syndrome. And the MT- ATP6 m.8993T>G (L156R) mutation causes neurogenic muscle weakness, ataxia and retinitis pigmentosa (NARP) and Leigh syndrome. More information regarding mtDNA mutation-related diseases can be found on <https://www.mitomap.org/foswiki/bin/view/MITOMAP/WebHome>, which could be taken as particular gene targets.

According to your suggestion, the following text has been added to the **Results** section in the revised manuscript (Page 19) to clarify the ability of InCasor to recognize single-base mutation sites far from the PAM sequence.: “To further examine InCasor's ability to recognize SNVs distant from the PAM sequence, we selected two loci with mutations at position 12 or 16 nt after PAM according to sequencing results or literature^{40,41}, respectively, and modified the crRNA to test its recognition by InCasor. Compared to fully matched crRNA and crRNA_{SM1}, the insertion of two discrete mismatches in the seed region (crRNA_{DM1}) and the insertion of 1 mismatch in the SNP locus (crRNA_{SM2}) both showed similarly improved discrimination factors (DFs) (for

mt.4769A>G, DF of crRNA_{SM2} was 4.1, DF of crRNA_{DM1} was 3.6; for mt.3916G>A, DF of crRNA_{SM2} was 2.4, DF of crRNA_{DM1} was 2.2) (**Supplementary Fig. 12**). Overall, engineering the crRNA contributed to InCasor's ability to sense SNV sites distant from the PAM site.” And the following text has been added to the **Discussion** section in the revised manuscript (Page 30): “Moreover, the crRNA engineering process substantially enhanced Cas12a’s ability for recognition of single-base mutations in non-PAM sequences.”

And Figure R14 has been amended as **Supplementary Fig. 12** in the revised **Supplementary Information** (Page 27).

- 40 Kopinski, P. K., Singh, L. N., Zhang, S., Lott, M. T. & Wallace, D. C. Mitochondrial DNA variation and cancer. *Nat Rev Cancer* **21**, 431-445, (2021).
41 McCormick, E. M. *et al.* Specifications of the ACMG/AMP standards and guidelines for mitochondrial DNA variant interpretation. *Human Mutation* **41**, 2028-2057, (2020).

have been added to **References** (Page 33) in the revised manuscript as marked with a gray background.

Comment 4:

Because of the number of panels per figure, some of the images are quite small at normal magnification. For example, individual mitochondria in Figure 4f are difficult to see, especially as the color merge between MT-Mito and Cy3-InCasor looks very similar to the separate panels. Zooming helps, but an inset with a single mitochondrion at higher magnification would be warranted.

Response 4:

Thank you for your suggestion. We have added an inset with a single mitochondrion at higher magnification in Fig. 4f. The results were shown as followed:

Figure R15. Confocal imaging of mitochondria extracted from MDA-MB-231 cells and HepG2 cells using InCasor^{ND4-2-MT} to detect mutant mtDNA (12084C>T). Scale bar: 25 μ m.

Based on the reviewer's suggestion, Figure R15 has been amended as new **Fig. 4 f** in the revised **manuscript** (Page 20) with yellow background.

Comment 5:

Sgc8- InCasor was highly localized to kidney and ND4-2-MT & ND5-MT-1 injected into the tail vein produced fluorescence outside the tumor region. The latter results seem to be explained as metastatic cells in the liver, but could these results indicate potential false-positives in other cells that weren't examined? Further, could false-positives arise if either the stability of the aptamer is compromised or via unrecognized gene-specific off-target effects with engineered crRNAs? Moreover, the Sgc8+ InCasor was not exclusively localized to the tumor. How could this be improved? These issues should be addressed in the Discussion.

Response 5: Thanks for your instructive suggestion. As described by the reviewers, there are two main potential sources of fluorescent signals generated in non-tumor regions: (1) tumor metastases carrying mtNDA mutations, and (2) potential false-positive signals, which could be generated by self-degradation of InCasor in the blood circulation or by genetic off-targeting of the CRISPR system.

The stability of InCasor *in vivo* is important for its signal-to-noise ratio. To test whether signals in non-tumor regions are false-positive signals, we firstly tested the stability of InCasor *in vitro* and *in vivo*, respectively. Specifically, single-stranded reporter (SSR), circular reporter (CLR), DNF/SSR, DNF/CLR, and InCasor (equal final concentration of reporter: 1 μ M) were respectively incubated with 90% serum for 0, 2,

4, 6, 12, 24, and 36 h at 37°C. After wards, the fluorescence of each reaction mixture was measured (λ_{ex} : 488 nm; λ_{em} : 520 nm). The results are shown as followed:

Figure R16. *In vitro* fluorescence assay to analyze the stability of InCasor in 90% serum over time. n=6, data show mean \pm SD.

As the result showed, compared to SSR which was rapidly degraded to produce the strongest fluorescent signal, CLR showed improved stability. Notably, both SSRs and CLR showed enhanced stability in 90% serum after loading onto DNF, suggesting that the dense spherical structure of DNF may protect the reporter from nuclease degradation. Similar to the DNF/CLR group, InCasor maintained a lower fluorescence signal in 90% serum, which highlights the potential of using such probe for *in vivo* detection of SNPs in mtDNA with high signal-to-noise ratio.

We then evaluated its nonspecific signaling over time in HepG2 tumor-bearing mice using small-animal *in vivo* imaging. Specifically, mice tail-vein injected with 200 μ L CLR, DNF/CLR, InCasors_{NT}, and InCasor_{Cy5} (equivalent amount of CLR: 10 nmoles), respectively. The whole-body images were obtained at 2, 4, 6, 8, 12, and 24 h using AniView100 by setting wavelength at excitation of 640 nm and emission of 710 nm. After 24 h of tail vein administration, mice were euthanized with CO₂ and the organs (heart, lung, liver, spleen, kidney) were removed and *ex vivo* imaged of Cy5 signal using AniView100. The section slides of the liver and kidney tissue were imaged by CLSM. The results are shown as followed:

Figure R17. Analyze the in vivo stability of InCasor. a. *In vivo* fluorescence signals of CLR, DNF/CLR, InCasor_{NT}, and InCasor_{Cy5} in HepG2 tumor-bearing mice over 24 h. **b.** Fluorescence imaging of harvested tumor and major organs (heart; liver; spleen; lung; and kidney) after 24 h post tail-vein injection of different probe. n=3, data show mean \pm SD. **c.** Semi-quantitative analysis of **b.** **d.** Representative images of the tumor, liver, and kidney tissue section in **d.** Scale bar: 50 μ m.

As shown in **Figure R17a**, in the free CLR group, a non-specific signal was detected at around 2 h and gradually increased with time, whereas DNF/CLR did not produce a similar phenomenon for 24 h, suggesting that the dense structure of DNF may protect the CLR from nuclease damage. InCasor_{NT}, which does not target mtDNA, also consistently maintained a lower Cy5 fluorescence signal compared with the CLR group and the InCasor_{Cy5}-positive control group. The *ex vivo* imaging results at 24 h post injection (**Figure R17b**) and corresponding quantification of fluorescence intensity (**Figure R17c**) were consistent with the results in **Figure R17a**. Notably, neither DNF/CLR nor InCasor_{NT} produced non-specific signals in major organs after 24 h injection. As revealed in **Figure R17d**, despite aggregating mainly in the kidney and liver, InCasor_{NT} did not produce non-specific Cy5 fluorescent signals within 24 h due to DNF protection. Collectively, these results indicate that InCasor demonstrates outstanding stability during blood circulation, which reduces the occurrence of false positive signals and lays the foundation for its accurate sensing of SNVs in mtDNA *in vivo*.

Further, could false-positives arise if either the stability of the aptamer is compromised or via unrecognized gene-specific off-target effects with engineered crRNAs?

Thanks for the question. We think the structure feature of DNF could minimize the possibility of aptamer compromise. Firstly, the DNF is stable under different conditions, including nuclease treatment and FBS incubation, which has been studied in many literature (*ACS Appl. Mater. Interfaces* 2019, 11, 46604–46613; *Biomaterials* 2020; 256, 120221; *Small* 2021, 17, 2104722; *Angew Chem Int Ed Engl.* 2020; 59(5):1897-1905.). Secondly, the DNF is consist of long repeats of template antisense sequence which contain aptamers, indicating that the RCA product contain lots of aptamers in it (*Chem. Soc. Rev.*, 2014, 43, 3324.). Based on this mechanism, even if the outer layer of Sgc8 aptamer is destroyed, the subsequently exposed inner layer of Sgc8 aptamer can still perform tumor-targeting effects.

For the possibility of unrecognized gene-specific off-target effects with engineered crRNAs, we have performed various processes to minimize the off-target effect. Firstly, we screen the whole genome sequence to make sure the target sequence didn't appear in other genomes. Secondly, the high-fidelity molecular recognition of Cas12a allows InCasor to recognize mutant mtDNA with high specificity without affecting wild-type mtDNA or other loci, which ensures its detection specificity. Each engineered crRNA used in this study was carefully designed to specificity recognize mutant mtDNA (**Fig. 4** and **Supplementary Fig. 11**). Besides, the whole probe system was designed to specifically targeting mitochondria, which decrease the possibility of nucleus off-target effect. As shown in **Supplementary Fig. 7**, the result of whole-genome sequencing (WGS) suggested that no obvious functional off-target gene editing was found in InCasor^{ND4} treated cells. Finally, all these designs work together to minimize the off-target effect.

Moreover, the Sgc8+ InCasor was not exclusively localized to the tumor. How could this be improved?

Complete localization to tumor tissue has been a long-standing challenge for nano-

delivery systems. *In vivo* fluorescence imaging showed that the Sgc8+ InCasor was predominantly distributed in the tumor, kidney, and liver (**Supplementary Fig. 13**), which was determined by a complex interplay of factors encompassing size, potential, nanostructure, and the tumor-targeting motifs inherent in the system's design. It is possible to improve the tumor targeting ability of probes by optimizing the synthesis conditions or modification with other motifs with higher targeting efficiency.

In addition, biomimetic systems that utilize cell's natural abilities (stealthies, increased agility/flexibility, prolonged circulation) recently attracted much attention in tumor-targeted drug delivery. Circulatory cell types such as erythrocytes, monocytes, macrophages, and even circulatory bacteria function as innate "moving gears," orchestrating the transport of cargo to the tumor site. These innovative hitchhiking strategies may be novel avenues for achieving efficient tumor delivery of nanoprobes.

Based on the reviewer's suggestion, the following text has been amended to the **Results** section (Page 6 and 24) in the revised manuscript with yellow background:

"The reporter cyclization strategy and the dense spherical structure of DNF protected the reporters from nuclease degradation, which improved the stability of InCasor in 90% serum (**Supplementary Fig. 1c**)." and "The *in vivo* fluorescence imaging and *ex vivo* organ fluorescence imaging was also performed to analyze the *in vivo* stability of InCasor (**Supplementary Fig. 16**). Due to the integrated nanostructure design, InCasor showed outstanding stability during blood circulation, which reduces the occurrence of false positive signals and lays the foundation for its accurate sensing of SNVs in mtDNA *in vivo*." And Figure R15 has been amended as **Supplementary Fig. 1c**, Figure R16 has been amended as **Supplementary Fig. 16** in the revised **Supplementary**

Information (Page 11 and 33).

Moreover, the following text has been amended to the **Discussion** section (Page 29) in the revised manuscript with yellow background: "Due to the limited copy number of mutant mtDNA in cells, a high signal amplification strategy is needed. In this work, using Mg²⁺ to enhance the trans-cleavage activity of Cas12a in live eukaryotic cells, we can efficiently report mtDNA mutations in live cells and *in vivo*. The integrated design of InCasor further leads to high local concentrations of Cas12a/crRNA and reporters

inside mitochondria, providing high sensitivity and signal-to-noise ratio for mtDNA analysis. The use of nanocarriers and cyclized reporter also greatly enhanced the probe stability *in vivo*, reducing the false-positive signals. Moreover, the crRNA engineering process substantially enhanced Cas12a's ability for recognition of SNVs in non-PAM sequences.”

Comment 6:

Line 270: Remove “Single nucleotide variation”; defined in line 24

Response 6:

Thanks for your comment. According to the reviewer's suggestion, we have removed “Single nucleotide variation” in line 330 in the revised **manuscript** (Page 17) and carefully examined the entire manuscript to avoid similar mistakes:

“As Cas12a holds the potential for identifying SNV^{29,35}, we were interested in determining whether the InCasor system could be used to discriminate SNV in mtDNA in live cells.”

REVIEWERS' COMMENTS

Reviewer #1 (Remarks to the Author):

The authors soundly responded to reviewers' comments, addressing most of the questions and concerns.

Reviewer #2 (Remarks to the Author):

The authors have answered all queries satisfactory. I have not further questions.

I look forward to the publication so that we can use some of their methodology for our mitochondrial research.

I thank the Editor to provide me with opportunity to review the manuscript.

Reviewer #3 (Remarks to the Author):

The authors have carefully addressed each of my original concerns. They have added new data, revised the Methods, Results, and Discussion, and expanded the Introduction and References. The results are interesting and potentially impactful for using InCasor as a detection platform for single nucleotide variants in mitochondria and beyond. I have no further suggestions.